# Practical Evaluation of Out-of-Distribution Detection Methods for Image Classification

## Abstract

We reconsider the evaluation of OOD detection methods for image recognition. Although many studies have been conducted so far to build better OOD detection methods, most of them follow Hendrycks and Gimpel's work for the method of experimental evaluation. While the unified evaluation method is necessary for a fair comparison, there is a question of if its choice of tasks and datasets reflect real-world applications and if the evaluation results can generalize to other OOD detection application scenarios. In this paper, we experimentally evaluate the performance of representative OOD detection methods for three scenarios, i.e., irrelevant input detection, novel class detection, and domain shift detection, on various datasets and classification tasks. The results show that differences in scenarios and datasets alter the relative performance among the methods. Our results can also be used as a guide for practitioners for the selection of OOD detection methods.

## 1 Introduction

Despite their high performance on various visual recognition tasks, convolutional neural networks (CNNs) often show unpredictable behaviors against out-of-distribution (OOD) inputs, i.e., those sampled from a different distribution from the training data. For instance, CNNs often classify irrelevant images to one of the known classes with high confidence. A visual recognition system should desirably be equipped with an ability to detect such OOD inputs upon its real-world deployment.

There are many studies of OOD detection that are based on diverse motivations and purposes. However, as far as the recent studies targeted at visual recognition are concerned, most of them follow the work of Hendrycks & Gimpel (2017), which provides a formal problem statement of OOD detection and an experimental procedure to evaluate the performance of methods. Employing this procedure, the recent studies focus mainly on increasing detection accuracy, where the performance is measured using the same datasets.

On the one hand, the employment of the experimental procedure has arguably bought about the rapid progress of research in a short period. On the other hand, little attention has been paid to how well the employed procedure models real-world problems and applications. They are diverse in purposes and domains, which obviously cannot be covered by the single problem setting with a narrow range of datasets.

In this study, to address this issue, we consider multiple, more realistic scenarios of the application of OOD detection, and then experimentally compare the representative methods. To be specific, we consider the three scenarios: *detection of irrelevant inputs*, *detection of novel class inputs*, and *detection of domain shift*. The first two scenarios differ in the closeness between ID samples and OOD samples.

Unlike the first two, domain shift detection is not precisely OOD detection. Nonetheless, it is the same as the other two in that what we want is to judge if the model can make a meaningful inference for a novel input. In other words, we can generalize OOD detection to the problem of judging this. Then, the above three scenarios are naturally fallen into the same group of problems, and it becomes natural to consider applying OOD detection methods to the third scenario. It is noteworthy that domain shift detection has been poorly studied in the community. Despite many demands from practitioners, there is no established method in the context of deep learning for image classification.

Based on the above generalization of OOD detection, we propose a meta-approach in which any OOD detection method can be used as its component.

For each of these three scenarios, we compare the following methods: the confidence-based baseline (Hendrycks & Gimpel, 2017), MC dropout (Gal & Ghahramani, 2016), ODIN (Liang et al., 2017), cosine similarity (Techapanurak et al., 2019; Hsu et al., 2020), and the Mahalanobis detector (Lee et al., 2018). Domain shift detection is studied in (Elsahar & Gallé, 2019) with natural language processing tasks, where proxy-A distance (PAD) is reported to perform the best; thus we test it in our experiments.

As for choosing the compared methods, we follow the argument shared by many recent studies (Shafaei et al., 2019; Techapanurak et al., 2019; Yu & Aizawa, 2019; Yu et al., 2020; Hsu et al., 2020) that OOD detection methods should not assume the availability of explicit OOD samples at training time. Although this may sound obvious considering the nature of OOD, some of the recent methods (e.g., Liang et al. (2017); Lee et al. (2018)) use a certain amount of OOD samples as *validation* data to determine their hyperparameters. The recent studies, (Shafaei et al., 2019; Techapanurak et al., 2019), show that these methods do perform poorly when encountering OOD inputs sampled from a different distribution from the assumed one at test time. Thus, for ODIN and the Mahalanobis detector, we employ their variants (Hsu et al., 2020; Lee et al., 2018) that can work without OOD samples. The other compared methods do not need OOD samples.

The contribution of this study are summarized as follows. i) Listing three problems that practitioners frequently encounter, we evaluate the existing OOD detection methods on each of them. ii) We show a practical approach to domain shift detection that is applicable to CNNs for image classification. iii) We show experimental evaluation of representative OOD detection methods on these problems, revealing each method's effectiveness and ineffectiveness in each scenario.

## 2 PROBLEMS AND METHODS

### 2.1 PRACTICAL SCENARIOS OF OOD DETECTION

We consider image recognition tasks in which a CNN classifies a single image $x$ into one of $C$ known classes. The CNN is trained using pairs of $x$ and its label, and $x$ is sampled according to $x \sim p(x)$. At test time, it will encounter an unseen input $x$, which is usually from $p(x)$ but is sometimes from $p'(x)$, a different, unknown distribution. In this study, we consider the following three scenarios.

**Detecting Irrelevant Inputs**   The new input $x$ does not belong to any of the known classes and is out of concern. Suppose we want to build a smartphone app that recognizes dog breeds. We train a CNN on a dataset containing various dog images, enabling it to perform the task with reasonable accuracy. We then point the smartphone to a sofa and shoot its image, feeding it to our classifier. It could classify the image as a *Bull Terrier* with high confidence. Naturally, we want to avoid this by detecting the irrelevance of $x$. Most studies of OOD detection assumes this scenario for evaluation.

**Detecting Novel Classes**   The input $x$ belongs to a novel class, which differs from any of $C$ known classes, and furthermore, we want our CNN to learn to classify it later, e.g., after additional training. For instance, suppose we are building a system that recognizes insects in the wild, with an ambition to make it cover all the insects on the earth. Further, suppose an image of one of the endangered (and thus rare) insects is inputted to the system while operating it. If we can detect it as a novel class, we would be able to update the system in several ways. The problem is the same as the first scenario in that we want to detect whether $x \sim p(x)$ or not. The difference is that $x$ is more similar to samples of the learned classes, or equivalently, $p'(x)$ is more close to $p(x)$, arguably making the detection more difficult. Note that in this study, we don't consider distinguishing whether $x$ is an irrelevant input or a novel class input, for the sake of simplicity. We left it for a future study.

**Detecting Domain Shift**   The input $x$ belongs to one of $C$ known classes, but its underlying distribution is $p'(x)$, not $p(x)$. We are especially interested in the case where a distributional shift $p(x) \rightarrow p'(x)$ occurs either suddenly or gradually while running a system for the long term. Our CNN may or may not generalize beyond this shift to $p'(x)$. Thus, we want to detect if it does not. If we can do this, we would take some actions, such as re-training the network with new training

data (Elsahar & Gallé, 2019). We consider the case where no information is available other than the incoming inputs $x's$.

A good example is a surveillance system using a camera deployed outdoor. Let us assume the images' quality deteriorates after some time since its deployment, for instance, due to the camera's aging. Then, the latest images will follow a different distribution from that of the training data. Unlike the above two cases where we have to decide for a single input, we can use multiple inputs; we should, especially when the quality of input images deteriorate gradually as time goes.

The problem here has three differences from the above two scenarios. First, the input is a valid sample belonging to a known class, neither an irrelevant sample nor a novel class sample. Second, we are basically interested in the accuracy of our CNN with the latest input(s) and not in whether $x \sim p(x)$ or $p'(x)$. Third, as mentioned above, we can use multiple inputs $\{x_i\}_{i=1,...,n}$ for the judgment.

Additional remarks on this scenario. Assuming a temporal sequence of inputs, the distributional shift is also called *concept drift* (Gama et al., 2014). It includes several different subproblems, and the one considered here is called *virtual* concept drift in its terminology. Mathematically, concept drift occurs when $p(x, y)$ changes with time. It is called *virtual* when $p(x)$ changes while $p(y|x)$ does not change. Intuitively, this is the case where the classes (i.e., concept) remain the same but $p(x)$ changes, demanding the classifier to deal with inputs drawn from $p'(x)$. Then, we are usually interested in predicting if $x$ lies in a region of the data space for which our classifier is well trained and can correctly classify it. If not, we might want to retrain our classifier using additional data or invoke unsupervised domain adaptation methods (Ganin & Lempitsky, 2015; Tzeng et al., 2017).

## 2.2 COMPARED METHODS

We select five representative OOD detection methods that do not use real OOD samples to be encountered at test time.

**Baseline: Max-softmax**  Hendrycks & Gimpel (2017) showed that the maximum of the softmax outputs, or confidence, can be used to detect OOD inputs. We use it as the score of an input being in-distribution (ID). We will refer to this method as *Baseline*. It is well known that the confidence can be calibrated using temperature to better represent classification accuracy (Guo et al., 2017; Li & Hoiem, 2020). We also evaluate this calibrated confidence, which will be referred to as *Calib*.

**MC Dropout**  The confidence (i.e., the max-softmax) is also thought of as a measure of uncertainty of prediction, but it captures only aleatoric uncertainty (Hüllermeier & Waegeman, 2019). Bayesian neural networks (BNNs) can also take epistemic uncertainty into account, which is theoretically more relevant to OOD detection. MC (Monte-Carlo) dropout (Gal & Ghahramani, 2016) is an approximation of BNNs that is computationally more efficient than an ensemble of networks (Lakshminarayanan et al., 2017). To be specific, using dropout (Srivastava et al., 2014) at test time provides multiple prediction samples, from which the average of their max-softmax values is calculated and used as ID score.

**Cosine Similarity**  It is recently shown in Techapanurak et al. (2019); Hsu et al. (2020) that using scaled cosine similarities at the last layer of a CNN, similar to the angular softmax for metric learning, enables accurate OOD detection. To be specific, the method first computes cosine similarities between the feature vector of the final layer and class centers (or equivalently, normalized weight vectors for classes). They are multiplied with a scale and then normalized by softmax to obtain class scores. The scale, which is the inverse temperature, is predicted from the same feature vector. These computations are performed by a single layer replacing the last layer of a standard CNN. The maximum of the cosine similarities (without the scale) gives ID score. The method is free of hyperparameters for OOD detection. We will refer to it as *Cosine*.

**ODIN (with OOD-sample Free Extension)**  ODIN was proposed by Liang et al. (2017) to improve *Baseline* by perturbing an input $x \to x + \epsilon \cdot \text{sgn}(\delta x)$ in the direction $\delta x$ of maximally increasing the max-softmax and also by temperature scaling. Thus, there are two hyperparameters, the perturbation size $\epsilon$ and the temperature $T$. In Liang et al. (2017), they are chosen by assuming the availability of explicit OOD samples. Recently, Hsu et al. (2020) proposed to select $\epsilon \leftarrow \text{argmax}_\epsilon \sum y_\kappa(x + \epsilon \cdot \text{sgn}(\delta x))$, where $y_\kappa$ is the max-softmax and the summation is taken over ID samples in the

validation set. As for the temperature, they set $T = 1000$. ID score is given by $y_\kappa(x + \epsilon \cdot \text{sgn}(\delta x))$. To distinguish from the original ODIN, we refer to this as ODIN*.

**Mahalanobis Detector**   The above three methods are based on the confidence. Another approach is to formulate the problem as unsupervised anomaly detection. Lee et al. (2018) proposed to model the distribution of intermediate layer's activation by a Gaussian distribution for each class but with a shared covariance matrix among the classes. Given an input, the Mahalanobis distance concerning the predicted class is calculated at each layer. A score for OOD is given by the weighted sum of those calculated at different layers. The weights are predicted by logistic regression, which is determined by assuming the availability of OOD samples. To be free from the assumption, another method is suggested that generates adversarial examples from ID samples and regard them as OOD samples. It is also reported in (Hsu et al., 2020) that setting all the weights to one works reasonably well. We evaluate the last two methods that do not need OOD samples. Although the original method optionally uses input perturbation similar to ODIN, we do not use it because our experiments show that its improvement is very small despite its high computational cost.

**Effects of Fine-tuning a Pre-trained Network**   It has been well known that fine-tuning a pre-trained network on a downstream task improves its prediction accuracy, especially when a small amount of training data is available. It was pointed out in (He et al., 2019) that the improvement is little when there is sufficient training data. Hendrycks et al. (2019) then show that even in that case, using a pre-trained network helps increase the overall robustness of the inference. It includes improved OOD detection performance, in addition to robustness to adversarial attacks, better calibration of confidence, robustness to covariate shift. However, their experimental validation is performed only on a single configuration with a few datasets. It remains unclear if the improvement can generalize to a broader range of purposes and settings that may differ in image size, the number of training samples, and ID/OOD combinations.

## 3   EXPERIMENTAL RESULTS

We use Resnet-50 (He et al., 2016) for a base network. We use it as is for *Baseline*, *ODIN**, and *Mahalanobis*, which share the same networks with the same weights, which will be referred to as *Standard*. We apply dropout to the last fully-connected layer with $p = 0.5$ and draw ten samples for *MC dropout*. We modify the last layer and the loss function for *Cosine*, following Techapanurak et al. (2019). We use the ImageNet pre-trained model provided by the Torchvision libraryfor their pre-trained models. We employ AUROC to evaluate OOD detection performance with the first two scenarios, following previous studies.

### 3.1   DETECTION OF IRRELEVANT INPUTS

We employ the following five tasks and datasets: dog breed recognition (120 classes and 10,222 images; Khosla et al. (2011)), plant seeding classification (12 classes and 5,544 images; Giselsson et al. (2017)), Food-101 (101 classes and 101,000 images; Bossard et al. (2014)), CUB-200 (200 classes and 11,788 images; Welinder et al. (2010)), and Stanford Cars (196 classes and 16,185 images; Krause et al. (2013)). These datasets

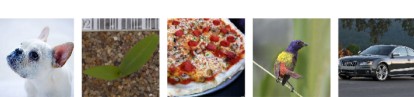

Figure 1: Example images for the five datasets.

will be referred to as *Dog*, *Plant*, *Food*, *Bird*, and *Cars*. They are diverse in terms of image contents, the number of classes, difficulty of tasks (e.g., fine-grained/coarse-grained), etc. Choosing one of the five as ID and training a network on it, we regard each of the other four as OOD, measuring the OOD detection performance of each method on the $5 \times 4$ ID-OOD combination. We train each network for three times to measure the average and standard deviation for each configuration. Table 1 shows the accuracy of the five datasets/tasks for the three networks (i.e., *Standard*, *MC dropout*, and *Cosine*) trained from scratch and fine-tuned from a pre-trained model, respectively. It is seen that there is large gap between training-from-scratch and fine-tuning a pre-trained model for the datasets with fewer training samples.

Figure 2 shows the average AUROC of the compared OOD detection methods for each ID dataset over the four OOD datasets and three trials for each. The error bars indicate the minimum and

Table 1: Classification accuracy (mean and standard deviation in parenthesis) of the three networks on the five datasets/tasks.

| Dataset | Train-from-scratch | | | Fine-tuning | | |
| | *Standard* | *MC dropout* | *Cosine* | *Standard* | *MC dropout* | *Cosine* |
|---|---|---|---|---|---|---|
| *Dog* | 26.7(3.4) | 28.9(2.8) | 36.2(1.4) | 79.4(0.1) | 79.3(0.3) | 78.5(0.3) |
| *Plant* | 94.1(0.4) | 94.7(0.2) | 95.8(0.9) | 95.2(0.6) | 95.5(0.5) | 92.7(2.6) |
| *Food* | 75.5(1.0) | 76.4(0.2) | 76.6(0.1) | 80.5(0.0) | 80.7(0.1) | 79.2(0.1) |
| *Bird* | 24.7(0.9) | 28.5(0.6) | 31.3(2.4) | 71.9(0.3) | 72.4(0.4) | 70.1(0.3) |
| *Car* | 18.2(3.8) | 22.0(1.6) | 36.0(6.2) | 77.6(0.3) | 77.7(0.3) | 73.7(0.6) |

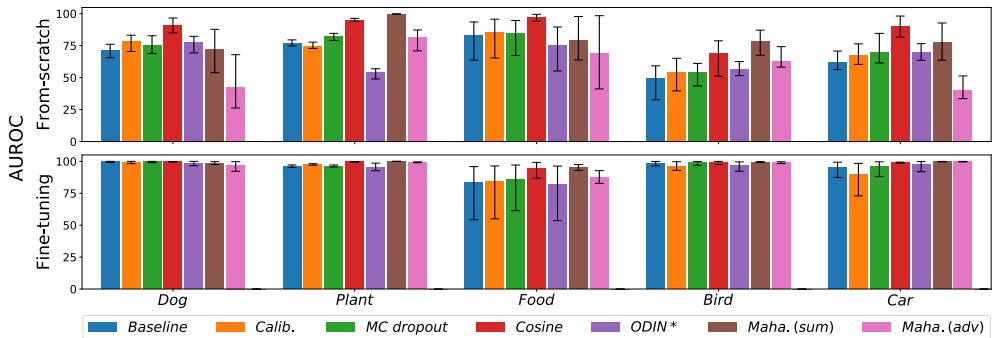

Figure 2: OOD detection performance of the compared methods. '*Dog*' indicates their performance when *Dog* is ID and all the other four datasets are OOD, etc. Each bar shows the average AUROC of a method, and the error bar indicates its minimum and maximum values. **Upper:** The networks are trained from scratch. **Lower:** Pre-trained models are fine-tuned.

maximum of AUROC. The full results for each of the twenty ID-OOD pairs are reported in Tables 5 and 6 in Appendix A. The upper row of Fig. 2 shows the results with the networks trained from scratch. It is seen that the ranking of the compared methods are mostly similar for different ID datasets. For the five datasets, *Cosine* is consistently among the top group; *Mahalanobis* will be ranked next, since it performs mediocre for *Dog* and *Food*. For the tasks with low classification accuracy, *Dog*, *Bird*, and *Car*, as shown in Table 1, the OOD detection accuracy tends to be also low; however, there is no tendency in the ranking of the OOD detection methods depending on the ID classification accuracy.

The lower row of Fig. 2 shows the results with the fine-tuned networks. It is first observed for any dataset and method that the OOD detection accuracy is significantly higher than the networks trained from scratch. This reinforces the argument made by Hendrycks et al. (2019) that the use of pre-trained networks improves OOD detection performance. Furthermore, the performance increase is a lot higher for several cases than reported in their experiments that use CIFAR-10/100 and Tiny ImageNet (Deng et al., 2009). The detection accuracy is pushed to a near-maximum for each case. Thus, there is only a little difference among the methods; *Cosine* and *Mahalanobis*(sum) shows slightly better performance for some datasets.

## 3.2 DETECTION OF NOVEL CLASSES

We conducted two experiments with different datasets. The first experiment uses the Oxford-IIIT Pet dataset (Parkhi et al., 2012), consisting of 25 dog breeds and 12 cat breeds. We use only the dog breeds and split them into 20 and 5 breeds. We then train each network on the first 20 dog breeds using the standard train/test splits per class. The remaining five breeds (i.e., *Scottish Terrier, Shiba Inu, Staffordshire Bull Terrier, Wheaten Terrier, Yorkshire Terrier*) are treated as OOD. It should be noted that the ImageNet dataset contains 118 dog breeds, some of which overlap with them. We intentionally leave this overlap to simulate a similar situation that could occur in practice. In the second experiment, we use the Food-101 dataset. We remove eight classes[1] contained in the

---

[1] Apple Pie, Breakfast Burrito, Chocolate Mousse, Gaucamole, Hamburger, Hot Dog, Ice Cream, Pizza

Table 2: Novel class detection performance of the compared methods measured by AUROC.

| Method | *Dog* | | *Food-A* | |
|---|---|---|---|---|
| | From-scratch | Fine-tuning | From-scratch | Fine-tuning |
| *Baseline* | 61.1(0.8) | 88.7(1.0) | 82.5(0.1) | 84.6(0.2) |
| *Calib.* | 62.9(0.5) | 86.5(1.1) | 83.4(0.1) | 84.8(0.2) |
| *MC dropout* | 61.7(0.4) | 89.8(0.8) | 82.7(0.1) | 84.7(0.1) |
| *Cosine* | **68.8(1.3)** | **94.1(0.8)** | **83.7(0.1)** | **85.7(0.3)** |
| *ODIN** | 59.9(0.5) | 85.3(1.4) | 77.4(0.1) | 74.7(0.3) |
| *Maha. (sum)* | 52.3(1.2) | 78.7(1.4) | 51.3(0.1) | 61.9(0.5) |
| *Maha. (adv)* | 49.0(0.2) | 65.8(1.5) | 51.8(0.3) | 57.1(7.2) |

ImageNet dataset. We split the remaining 93 classes into 46 and 47 classes, called *Food-A* and *-B*, respectively. Each network is trained on *Food-A*. We split *Food-A* into 800/100/100 samples per class to form train/val/test sets. Treating *Food-B* as OOD, we evaluate the methods' performance.

Table 2 shows the methods' performance of detecting OOD samples (i.e., novel samples). In the table we separate the Mahalanobis detector and the others; the latter are all based on confidence or its variant, whereas Mahalanobis is not. The ranking of the methods is similar between the two experiments. *Cosine* attains the top performance for both of the two training methods. While this is similar to the results of irrelevant sample detection (Fig. 2), the gap to the second best group (*Baseline*, *Calib.*, and *MC dropout*) is much larger here; this is significant for training from scratch. Another difference is that neither variant of *Mahalanobis* performs well; they are even worse than *Baseline*. This will be attributable to the similarity between ID and OOD samples here. The classification accuracy of the original tasks, *Dog* and *Food-A* are given in Table 7 in Appendix B.

## 3.3 DETECTION OF DOMAIN SHIFT

### 3.3.1 PROBLEM FORMULATION

Given a network trained on a dataset $\mathcal{D}_s$, we wish to estimate its classification error on a different dataset $\mathcal{D}_t$. In practice, a meta-system monitoring the network estimates the classification error on each of the incoming datasets $\mathcal{D}_t^{(1)}, \mathcal{D}_t^{(2)}, \cdots$, which are chosen from the incoming data stream. It issues an alert if the predicted error for the latest $\mathcal{D}_t^{(T)}$ is higher than the pre-fixed target.

We use an OOD score $S$ for this purpose. To be specific, given $\mathcal{D}_t = \{x_i\}_{i=1,\dots,n}$, we calculate an average of the score $\overline{S} = \sum_i^n S_i/n$, where $S_i$ is the OOD score for $x_i$; note that an OOD score is simply given by a negative ID score. We want to use $\overline{S}$ to predict the classification error $\overline{err} = \sum_{i=1}^n 1(y_i = t_i)/n$, where $y$ and $t$ are a prediction and the true label, respectively. Following Elsahar & Gallé (2019), we train a regressor $f$ to do this, as $\overline{err} \sim f(\overline{S})$. We assume multiple labeled datasets $\mathcal{D}_o$'s are available, each of which do not share inputs with $\mathcal{D}_s$ or $\mathcal{D}_t$. Choosing a two-layer MLP for $f$, we train it on $\mathcal{D}_o$'s plus $\mathcal{D}_s$. As they have labels, we can get the pair of $\overline{err}$ and $\overline{S}$ for each of them. Note that $\mathcal{D}_t$ does not have labels.

It is reported in Elsahar & Gallé (2019) that Proxy-A Distance (PAD) (Ben-David et al., 2007) performs well on several NLP tasks. Thus, we also test this method (rigorously, the one called PAD* in their paper) for comparisons. It first trains a binary classifier using portions of $\mathcal{D}_s$ and $\mathcal{D}_t$ to distinguish the two. Then, the classifier's accuracy is evaluated on the held-out samples of $\mathcal{D}_s$ and $\mathcal{D}_t$, which is used as a metric of the distance between their underlying distributions. Intuitively, the classification is easy when their distance is large, and vice versa. We train $f$ using $1 - $ (mean absolute error) for $S$ as in the previous work.

### 3.3.2 DOMAIN SHIFT BY IMAGE CORRUPTION

We first consider the case when the shift is caused by the deterioration of image quality. An example is a surveillance camera deployed in an outdoor environment. Its images are initially of high quality, but later their quality deteriorates gradually or suddenly due to some reason, e.g., dirt on the lens, failure of focus adjustment, seasonal/climate changes, etc. We want to detect it *if it affects classifi-*

Table 3: Errors of the predicted classification error by the compared methods.

| Method | Food-A (From-scratch) | | Food-A (Fine-tuning) | | ImageNet | |
|--------|--------|--------|--------|--------|--------|--------|
| | MAE | RMSE | MAE | RMSE | MAE | RMSE |
| Baseline | 15.8(3.0) | 20.5(3.7) | 6.4(1.3) | 7.9(1.6) | 4.6(0.8) | 6.3(1.0) |
| Calib. | 15.0(2.9) | 19.6(3.4) | 6.3(1.3) | 7.9(1.6) | 4.3(0.8) | 6.0(1.0) |
| MC dropout | 15.3(2.7) | 19.7(3.0) | **5.8(1.1)** | **7.2(1.4)** | 4.0(0.7) | 5.3(0.9) |
| Cosine | **6.6(1.3)** | **8.2(1.6)** | 6.1(1.6) | 7.5(2.2) | **3.8(0.9)** | **4.7(1.1)** |
| ODIN* | 14.7(1.9) | 17.4(2.2) | 8.9(1.3) | 10.8(1.4) | 9.1(0.8) | 12.3(1.2) |
| Maha. (sum) | 15.3(1.5) | 18.4(1.8) | 15.6(1.5) | 18.9(2.1) | 15.1(2.2) | 18.5(2.9) |
| Maha. (adv) | 14.3(1.5) | 17.5(2.0) | 19.1(15.8) | 24.1(27.6) | 16.1(1.7) | 19.6(2.6) |
| PAD | 16.3(1.5) | 19.2(1.9) | 17.5(1.3) | 20.5(1.6) | 11.0(1.1) | 12.9(1.2) |

*cation accuracy.* To simulate multiple types of image deterioration, we employ the method and code for generating image corruption developed by Hendrycks & Dietterich (2019). It can generate 19 types of image corruptions, each of which has five levels of severity.

We consider two classification datasets/tasks, *Food-A* (i.e., 46 selected classes from Food-101 as explained in Sec. 3.2) and ImageNet (the original 1,000 object classification). For *Food-A*, we first train each network on the training split, consisting only of the original images. We divide the test split into three sets, 1,533, 1,533, and 1,534 images, respectively. The first one is used for $\mathcal{D}_s$ as is (i.e., without corruption). We apply the image corruption method to the second and third sets. To be specific, splitting the 19 corruption types into 6 and 13, we apply the 6 corruptions to the second set to make $\mathcal{D}_o$'s, and the 13 corruptions to the last to make $\mathcal{D}_t$'s. As each corruption has five severity levels, there are $30(= 6 \times 5)$ $\mathcal{D}_o$'s and $65(= 13 \times 5)$ $\mathcal{D}_t$'s. The former is used for training $f$ (precisely, 20 are used for training and 10 are for validation), and the latter is for evaluating $f$.

For ImageNet, we choose 5,000, 2,000, and 5,000 images from the validation split without overlap. We use them to make $\mathcal{D}_s$, $\mathcal{D}_o$'s, and $\mathcal{D}_t$'s, respectively. As with *Food-A*, we apply the 6 and 13 types of corruption to the second and third sets, making 30 $\mathcal{D}_o$'s and 65 $\mathcal{D}_t$'s, respectively.

For the evaluation of $f$, we calculate *mean absolute error (MAE)* and *root mean squared error (RMSE)* of the predicted $\overline{\mathrm{err}}$ over the 65 $\mathcal{D}_t$'s. We repeat this for 20 times with different splits of image corruptions ($19 \rightarrow 6 + 13$), reporting their mean and standard deviation.

Table 3 shows the results for *Food-A* and ImageNet. (The accuracies of the original classification tasks of *Food-A* and ImageNet are reported in Table 7 and Table 10 in Appendix B and C.) It is seen for both datasets that *Cosine* achieves the top-level accuracy irrespective of the training methods. For *Food-A*, using a pre-trained network boosts the performance for the confidence-based methods (i.e., from *Baseline* to *ODIN\**), resulting in that *MC dropout* performs the best; *Cosine* attains almost the same accuracy. On the other hand, *Mahalanobis* and *PAD* do not perform well regardless of the datasets and training methods. This well demonstrates the difference between detecting the distributional shift $p(x) \rightarrow p'(x)$ and detecting the deterioration of classification accuracy. We show scatter plots of $\overline{S}$ vs. $\overline{\mathrm{err}}$ in Fig. 4 and 5 in Appendix C, which provides a similar, or even clearer, observation.

### 3.3.3 OFFICE-31

To study another type of domain shift, we employ the Office-31 dataset (Saenko et al., 2010), which is popular in the study of domain adaptation. The dataset consists of three subsets, Amazon, DSLR, and Webcam, which share the same 31 classes and are collected from different domains. We train our CNNs on Amazon and evaluate the compared methods in terms of prediction accuracy of classification errors for samples in DSLR and Webcam. The classification accuracy of the CNNs on Amazon is provided in Table 11 in Appendix D.

To obtain $\mathcal{D}_o$'s for training $f$, we employ the same image corruption methods as Sec. 3.3.2; we apply them to Amazon samples to create virtual domain-shifted samples. The effectiveness of modeling the true shifted data, i.e., DSLR and Webcam, with these samples is unknown and needs to be experimentally validated. If this works, it will be practically useful. Specifically, we split the test splits of Amazon containing 754 images evenly into two sets. We use one for $\mathcal{D}_s$ and the other for

Table 4: Errors of the predicted classification error by the compared methods on 50 sample subsets of DSLR and Webcam. The CNN is trained on Amazon and the regressor $f$ is trained using corrupted images of Amazon.

| Method | Train-from-scratch | | Fine-tuning | |
|---|---|---|---|---|
| | MAE | RMSE | MAE | RMSE |
| *Baseline* | 12.1(3.1) | 14.6(3.1) | 10.6(2.3) | 11.7(2.3) |
| *Calib.* | 9.5(3.2) | 11.5(3.4) | 9.7(2.3) | 10.8(2.3) |
| *MC dropout* | 8.0(1.9) | 10.1(2.7) | 9.3(1.7) | 10.5(1.7) |
| *Cosine* | **5.6(1.5)** | **6.8(1.7)** | 8.5(2.0) | 10.0(2.2) |
| *ODIN\** | 7.3(2.5) | 8.1(2.6) | 13.4(3.6) | 15.3(3.6) |
| *Maha. (sum)* | 18.8(4.7) | 20.7(3.9) | **7.9(2.0)** | **9.7(2.1)** |
| *Maha. (adv)* | 34.1(15.8) | 40.0(22.5) | 8.2(2.1) | 9.9(2.2) |
| *PAD* | 10.6(2.2) | 12.1(2.6) | 16.8(3.4) | 18.3(3.2) |

creating $\mathcal{D}_o$'s. We apply all the types of corruption, yielding $95(= 19 \times 5)$ $\mathcal{D}_o$'s. We then split them into those generated by four corruptions and those generated by the rest; the latter is used for training $f$, and the former is used for the validation. We iterate this for 20 times with different random splits of the corruption types, reporting the average over $20 \times 3$ trials, as there are three CNN models trained from different initial weights.

To evaluate each method (i.e., $f$ based on a OOD score), we split DSLR and Webcam into subsets containing 50 samples, yielding 18 $\mathcal{D}_t$'s in total. We apply $f$ to each of them, reporting the average error of predicting classification errors. Table 4 shows the results. It is observed that *Cosine* works well in both training methods. The two variants of *Mahalanobis* show good performance when using a pre-trained model, but this may be better considered a coincidence, as explained below. Figure 3 shows the scatter plots of OOD score vs. classification error for each method. The green dots indicate $D_o$'s, corrupted Amazon images used for training $f$, and the blue ones indicate $D_t$'s, subsets from DSLR and Webcam containing 50 samples each. For the method for which the green dots distribute with narrower spread, the regressor $f$ will yield more accurate results. Thus, it is seen from Fig. 3 that both *Mahalanobis*'s tend to have large spread, meaning that they could perform poorly depending on incoming domain-shifted data. *Cosine* and *MC dropout* have narrower spread, confirming their performance in Table 4. Other results for DSLR and Webcam subsets with a different number of samples are provided in Appendix D.

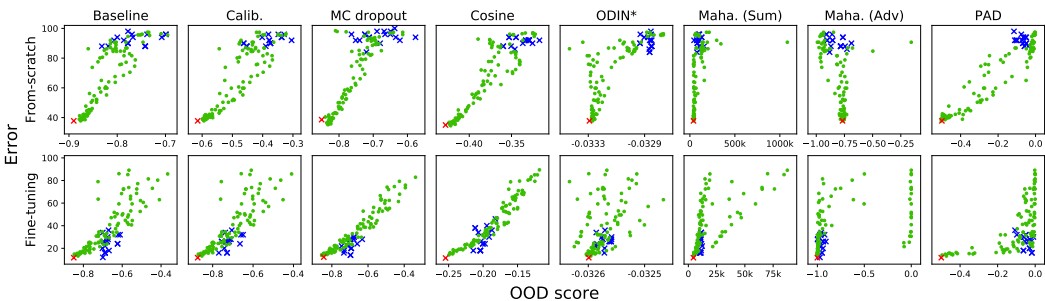

Figure 3: OOD score vs. the true classification error for $95(= 19 \times 5)$ $D_o$'s (corrupted Amazon images used for training the regressor $f$; in green), 18 $D_t$'s (subsets of DSLR and Webcam containing 50 samples each; in blue), and $D_s$ (original Amazon images; in red).

## 3.4 ANALYSES OF THE RESULTS

We can summarize our findings in the following. i) Using a pre-trained network has shown improvements in all the scenarios, confirming the report of Hendrycks et al. (2019). ii) The detector using cosine similarity consistently works well throughout the three scenarios. The method will be the first choice if it is acceptable to modify the network's final layer. iii) The Mahalanobis detector, a SOTA method, works well only for irrelevant input detection. This is not contradictory with the

previous reports, since they employ only this very scenario. The method fits a Gaussian distribution to ID samples belonging to each class and uses the same covariance matrix for all the classes. This strategy might work well on easy cases when incoming OOD samples are mapped distantly from the Gaussian distributions. However, such a simple modeling method will not work in more challenging cases. For instance, incoming OOD samples could be mapped near the ID distributions, as in novel class detection. In such cases, the ID sample distribution needs to be very precisely modeled, for which the assumption of Gaussian distributions with a single covariance matrix is inadequate. iv) Domain shift detection requires detecting classification accuracy deterioration, not detecting a distributional shift of inputs, contrary to its name. This theoretically favors the confidence-based methods; they (particularly MC dropout) indeed work well, when used with a pre-trained network. However, the Mahalanobis detector is more like an anomaly detection method, although its similarity with a softmax classifier is suggested in (Lee et al., 2018). An input sample for which the network can make a correct classification can be detected as an 'anomaly' by the Mahalanobis detector.

## 4  RELATED WORK

Many studies of OOD detection have been conducted so far, most of which are proposals of new methods; those not mentioned above include (Vyas et al., 2018; Yu & Aizawa, 2019; Sastry & Oore, 2020; Zisselman & Tamar, 2020; Yu et al., 2020). Experimental evaluation similar to our study but on the estimation of the uncertainty of prediction is provided in (Ovadia et al., 2019).

In (Hsu et al., 2020), the authors present a scheme for conceptually classifying domain shifts in two axes, semantic shift and non-semantic shift. Semantic shift (S) represents OOD samples coming from the distribution of an unseen class, and non-semantic shift (NS) represents to OOD samples coming from an unseen domain. Through the experiments using the DomainNet dataset (Peng et al., 2019), they conclude that OOD detection is more difficult in the order of S > NS > S+NS.

In this study, we classify the problems into three types from an application perspective. One might view this as somewhat arbitrary and vague. Unfortunately, Hsu et al.'s scheme does not provide help. For instance, according to their scheme, novel class detection is S, and domain shift is NS. However, it is unclear which to classify irrelevant detection between S and S+NS. Moreover, their conclusion (i.e., S > NS > S+NS) does not hold for our results; the difficulty depends on the closeness between classes and between domains. After all, we think that only applications can determine what constitutes domain and what constitutes classes. Further discussion will be left for a future study.

As mentioned earlier, the detection of domain shift in the context of deep learning has not been well studied in the community. The authors are not aware of a study for image classification and find only a few Elsahar & Gallé (2019) even when looking at other fields. On the other hand, there are a large number of studies of *domain adaptation* (DA); (Ganin & Lempitsky, 2015; Tzeng et al., 2017; Zhang et al., 2017; Toldo et al., 2020; Zou et al., 2019) to name a few. It is to make a model that has learned a task using the dataset of a particular domain adapt to work on data from a different domain. Researchers have been studied several problem settings, e.g., closed-set, partial, open-set, and boundless DA (Toldo et al., 2020). However, these studies all assume that the source and target domains are already known; no study considers the case where the domain of incoming inputs is unidentified. Thus, they do not provide a hint of how to detect domain shift.

## 5  SUMMARY AND CONCLUSION

In this paper, we first classified OOD detection into three scenarios from an application perspective, i.e., irrelevant input detection, novel class detection, and domain shift detection. We have presented a meta-approach to be used with any OOD detection method to domain shift detection, which has been poorly studied in the community. We have experimentally evaluated various OOD detection methods on these scenarios. The results show the effectiveness of the above approach to domain shift several as well as several findings such as which method works on which scenario.

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

# A  Detection of Irrelevant Inputs

## A.1  Additional Results

In our experiment for irrelevant input detection, using five datasets, we consider every pair of them, one for ID and the other for OOD. In the main paper, we reported only the average detection accuracy over four such pairs for an ID dataset. We report here the results for all the ID-OOD pairs. Tables 5 and 6 show the performance of the compared methods for training from scratch and for fine-tuning of a pre-trained network.

Table 5: The OOD detection performance (AUROC) for networks trained from scratch. D1=*Dog*, D2=*Plant*, D3=*Food*, D4=*Bird*, and D5=*Cat*.

| Method | OOD | In-Distribution | | | | |
| --- | --- | --- | --- | --- | --- | --- |
| | | D1 | D2 | D3 | D4 | D5 |
| *Baseline* | D1 | - | 78.1(1.1) | 88.9(0.7) | 59.2(1.0) | 61.4(3.9) |
| | D2 | 76.1(12.4) | - | 63.6(4.5) | 32.7(5.6) | 70.8(21.1) |
| | D3 | 65.5(4.2) | 75.9(4.6) | - | 54.1(1.2) | 56.3(4.4) |
| | D4 | 72.1(3.2) | 74.8(2.1) | 88.0(0.8) | - | 61.2(4.4) |
| | D5 | 73.4(2.4) | 79.5(5.4) | 93.5(1.0) | 53.0(0.9) | - |
| *Calib.* | D1 | - | 76.1(1.3) | 91.6(0.6) | 65.1(0.6) | 67.2(3.5) |
| | D2 | 83.2(12.2) | - | 65.3(5.3) | 39.6(3.9) | 76.4(20.3) |
| | D3 | 70.4(5.4) | 73.6(7.6) | - | 58.2(1.5) | 60.3(5.6) |
| | D4 | 78.9(3.4) | 72.8(1.9) | 90.8(0.8) | - | 66.5(4.5) |
| | D5 | 81.4(3.0) | 77.7(6.4) | 95.7(0.8) | 54.4(1.7) | - |
| *MC dropout* | D1 | - | 82.6(2.8) | 89.3(0.2) | 61.2(1.1) | 67.0(1.9) |
| | D2 | 82.5(9.9) | - | 67.4(7.7) | 43.8(7.4) | 84.4(8.8) |
| | D3 | 68.6(1.6) | 84.7(2.6) | - | 57.1(1.6) | 61.3(2.8) |
| | D4 | 73.7(1.6) | 79.0(2.0) | 88.3(0.4) | - | 67.3(1.4) |
| | D5 | 75.9(2.4) | 84.6(4.6) | 94.7(0.0) | 57.0(0.7) | - |
| *Cosine* | D1 | - | 93.9(2.2) | 96.9(0.2) | 74.8(0.5) | 89.7(2.3) |
| | D2 | 96.6(1.8) | - | 94.1(0.5) | 51.1(13.9) | 98.1(1.4) |
| | D3 | 85.0(1.4) | 94.4(2.5) | - | 78.8(1.9) | 81.7(4.7) |
| | D4 | 90.9(0.7) | 94.1(1.4) | 97.6(0.1) | - | 90.6(2.7) |
| | D5 | 92.5(0.5) | 96.3(0.9) | 99.5(0.0) | 73.6(2.5) | - |
| *ODIN\** | D1 | - | 55.8(16.6) | 74.2(0.9) | 54.5(2.1) | 63.5(2.1) |
| | D2 | 78.4(3.1) | - | 55.2(9.2) | 58.6(11.8) | 76.5(6.6) |
| | D3 | 69.3(6.1) | 54.9(21.9) | - | 62.5(3.0) | 68.7(2.0) |
| | D4 | 82.3(2.4) | 48.9(15.1) | 82.3(0.6) | - | 70.3(1.5) |
| | D5 | 81.2(3.8) | 57.0(18.1) | 89.6(0.7) | 51.6(2.7) | - |
| *Maha. (sum)* | D1 | - | 99.7(0.1) | 68.6(0.8) | 67.5(1.8) | 63.6(3.0) |
| | D2 | 87.8(3.4) | - | 97.8(0.6) | 75.7(4.5) | 92.7(5.9) |
| | D3 | 73.2(2.7) | 99.8(0.1) | - | 87.1(1.4) | 78.4(2.8) |
| | D4 | 75.9(1.2) | 99.7(0.1) | 87.2(0.8) | - | 76.2(2.0) |
| | D5 | 53.9(2.7) | 99.8(0.1) | 63.9(2.6) | 83.2(1.8) | - |
| *Maha. (adv)* | D1 | - | 80.9(10.8) | 65.6(3.1) | 59.8(3.6) | 35.2(8.2) |
| | D2 | 67.9(6.0) | - | 98.4(0.4) | 58.2(8.5) | 51.3(23.0) |
| | D3 | 41.0(4.2) | 87.0(8.2) | - | 59.2(3.5) | 39.5(9.7) |
| | D4 | 26.2(3.3) | 87.3(6.1) | 71.0(2.4) | - | 33.5(15.2) |
| | D5 | 35.5(6.6) | 70.9(19.4) | 41.1(6.6) | 74.2(9.7) | - |

Table 6: The OOD detection performance (AUROC) for fine-tuned networks from a pre-trained model. D1=*Dog*, D2=*Plant*, D3=*Food*, D4=*Bird*, and D5=*Cat*.

| Method | OOD | In-Distribution | | | | |
|---|---|---|---|---|---|---|
| | | D1 | D2 | D3 | D4 | D5 |
| *Baseline* | D1 | - | 96.3(0.4) | 91.4(0.5) | 96.4(0.6) | 99.3(0.5) |
| | D2 | 100.0(0.0) | - | 54.3(8.2) | 96.9(2.8) | 87.4(4.1) |
| | D3 | 99.7(0.0) | 96.1(0.7) | - | 99.2(0.2) | 97.1(0.6) |
| | D4 | 99.0(0.2) | 97.2(0.4) | 91.1(0.5) | - | 97.6(0.6) |
| | D5 | 100.0(0.0) | 95.2(1.5) | 95.9(1.2) | 99.8(0.1) | - |
| *Calib.* | D1 | - | 97.4(0.4) | 92.3(0.5) | 94.2(0.9) | 98.4(1.1) |
| | D2 | 100.0(0.0) | - | 55.0(8.3) | 92.9(6.0) | 73.0(6.2) |
| | D3 | 99.4(0.1) | 97.1(0.6) | - | 98.2(0.5) | 94.2(1.1) |
| | D4 | 98.1(0.5) | 98.1(0.3) | 91.8(0.5) | - | 94.7(1.1) |
| | D5 | 100.0(0.0) | 96.9(1.1) | 96.4(1.1) | 99.7(0.1) | - |
| *MC dropout* | D1 | - | 95.7(1.8) | 92.7(0.5) | 96.8(0.4) | 99.6(0.2) |
| | D2 | 100.0(0.0) | - | 61.4(7.3) | 99.3(0.4) | 88.0(7.1) |
| | D3 | 99.7(0.0) | 96.1(0.9) | - | 99.3(0.2) | 97.9(0.8) |
| | D4 | 98.9(0.1) | 97.2(0.7) | 91.5(0.7) | - | 98.7(0.5) |
| | D5 | 100.0(0.0) | 96.4(1.1) | 97.1(0.7) | 99.9(0.0) | - |
| *Cosine* | D1 | - | 99.3(0.5) | 96.3(0.0) | 97.5(0.2) | 99.4(0.0) |
| | D2 | 99.5(0.2) | - | 86.7(6.1) | 100.0(0.0) | 98.6(0.4) |
| | D3 | 99.5(0.1) | 99.5(0.2) | - | 99.4(0.1) | 99.0(0.3) |
| | D4 | 99.5(0.0) | 99.6(0.3) | 96.0(0.3) | - | 99.1(0.2) |
| | D5 | 99.8(0.0) | 99.8(0.1) | 99.2(0.2) | 99.6(0.0) | - |
| *ODIN** | D1 | - | 93.1(1.7) | 86.2(0.7) | 92.2(0.6) | 99.8(0.1) |
| | D2 | 99.6(0.2) | - | 53.5(16.5) | 98.0(0.9) | 91.8(2.8) |
| | D3 | 96.6(0.6) | 92.7(3.3) | - | 96.8(0.5) | 98.2(0.2) |
| | D4 | 97.2(0.6) | 95.7(1.3) | 90.0(0.3) | - | 99.6(0.1) |
| | D5 | 100.0(0.0) | 98.6(0.1) | 96.2(0.7) | 99.5(0.2) | - |
| *Maha. (sum)* | D1 | - | 100.0(0.0) | 93.2(0.5) | 99.0(0.1) | 99.6(0.0) |
| | D2 | 99.7(0.0) | - | 97.5(0.2) | 99.8(0.0) | 99.9(0.0) |
| | D3 | 98.9(0.0) | 100.0(0.0) | - | 99.3(0.1) | 99.8(0.0) |
| | D4 | 97.4(0.1) | 100.0(0.0) | 96.4(0.1) | - | 99.5(0.0) |
| | D5 | 98.1(0.0) | 100.0(0.0) | 92.8(0.4) | 99.1(0.0) | - |
| *Maha. (adv)* | D1 | - | 99.1(1.0) | 86.9(1.5) | 98.2(0.1) | 99.6(0.0) |
| | D2 | 99.7(0.0) | - | 82.7(6.2) | 99.7(0.0) | 100.0(0.0) |
| | D3 | 98.5(0.1) | 99.0(1.0) | - | 98.3(0.2) | 99.8(0.0) |
| | D4 | 92.2(1.1) | 98.9(1.2) | 92.7(1.1) | - | 99.5(0.0) |
| | D5 | 98.1(0.1) | 99.6(0.4) | 88.1(1.3) | 98.9(0.1) | - |

# B    DETECTION OF NOVEL CLASSES

## B.1    CLASSIFICATION ACCURACY OF THE BASE TASKS

In our experiments for novel class detection, we employ two datasets, *Dog* and *Food-A*. Table 7 shows the classification accuracy for each of them. It is seen that for *Dog*, using a pre-trained model boosts the accuracy. There is a tendency similar to that seen in Table 1, that *Cosine* outperforms others in training from scratch. For *Food-A*, using a pre-trained model shows only modest improvement due to the availability of a sufficient number of samples.

Table 7: Classification accuracy for the two tasks, *Dog* (20 dog breeds classification) and *Food-A* (46 food class classification), for which novel class detection is examined.

| Dataset | Train-from-scratch | | | Fine-tuning | | |
|---|---|---|---|---|---|---|
| | *Standard* | *MC dropout* | *Cosine* | *Standard* | *MC dropout* | *Cosine* |
| *Dog* | 51.9(0.8) | 51.1(0.8) | 64.2(1.4) | 95.7(0.1) | 95.6(0.2) | 96.0(0.1) |
| *Food-A* | 83.4(0.3) | 84.0(0.2) | 83.6(0.1) | 87.5(0.2) | 87.5(0.2) | 86.0(0.1) |

## B.2    ADDITIONAL RESULTS

In one of the experiments explained in Sec. 3.2, we use only dog classes from the Oxford-IIIT Pet dataset. We show here additional results obtained when using cat classes. Choosing nine from 12 cat breeds contained in the dataset, we train the networks on classification of these nine breeds and test novel class detection using the remaining three breed classes. In another experiment, we use *Food-A* for ID and *Food-B* for OOD. We report here the results for the reverse configuration. Table 8 shows the classification accuracy of the new tasks. Table 9 shows the performance of the compared methods on the novel class detection. A similar observation to the experiments of Sec. 3.2 can be made.

Table 8: Classification accuracy for *Cat* (9 cat breed classification) and *Food-B* (47 food class classification).

| Dataset | Random Init. | | | Fine-Tuning | | |
|---|---|---|---|---|---|---|
| | *Standard* | *MC dropout* | *Cosine* | *Standard* | *MC dropout* | *Cosine* |
| *Cat* | 60.9(0.8) | 57.8(1.2) | 64.1(0.9) | 89.5(0.3) | 88.8(0.3) | 88.9(0.5) |
| *Food-B* | 83.6(0.3) | 84.3(0.4) | 83.7(0.3) | 87.6(0.2) | 87.4(0.1) | 86.2(0.1) |

Table 9: Novel class detection performance (AUROC) of the compared methods. The OOD samples for *Cat* and *Food-B* are the held-out 3 cat breeds and *Food-A*, respectively.

| Method | *Cat* | | *Food-B* | |
|---|---|---|---|---|
| | From-scratch. | Fine-Tune | From-scratch | Fine-Tune |
| *Baseline* | 57.7(0.1) | 72.1(1.9) | 81.5(0.2) | 84.2(0.2) |
| *Calib.* | 58.6(0.4) | 70.9(2.0) | **82.5(0.2)** | 84.5(0.2) |
| *MC dropout* | 57.6(1.3) | 72.7(1.1) | 82.0(0.2) | 84.5(0.3) |
| *Cosine* | **63.6(1.1)** | **73.7(2.1)** | 81.7(0.2) | **84.8(0.3)** |
| *ODIN** | 52.3(0.7) | 72.6(0.7) | 74.4(0.7) | 73.6(0.3) |
| *Maha. (sum)* | 43.9(0.5) | 62.2(0.4) | 51.8(0.2) | 64.2(0.1) |
| *Maha. (adv)* | 50.0(3.0) | 61.1(2.1) | 52.6(0.4) | 52.8(9.8) |

## C    DETECTION OF DOMAIN SHIFT (IMAGE CORRUPTION)

### C.1    CLASSIFICATION ACCURACY ON IMAGENET

Table 10 shows the accuracy of the three networks used by the compared OOD detection methods for 1,000 class classification of the ImageNet dataset. We use center-cropping at test time. The cosine network shows lower classification accuracy here.

Table 10: Classification accuracy on ImageNet for each network.

| Dataset | *Standard* | *MC dropout* | *Cosine* |
|---------|-----------|-------------|---------|
| ImageNet | 74.6(0.1) | 75.5(0.3) | 72.4(0.1) |

### C.2    SCATTER PLOTS OF OOD SCORE VS. CLASSIFICATION ERROR

In Sec. 3.3.2, we showed experimental results of domain shift detection using *Food-A*. Given a set $\mathcal{D}_t$ of samples, each of the compared methods calculates an OOD score $\overline{S}$ for it, from which the average classification error $\overline{\text{err}}$ over samples from $\mathcal{D}_t$ is predicted. Figure 4 shows scatter plots showing the relation between the OOD score $\overline{S}$ and the true classification error for a number of datasets (i.e., $\mathcal{D}_t$'s). We have $95(=19 \times 5)$ such datasets, each containing images undergoing one of the combinations of 19 image corruptions and 5 severity levels. The method with a narrower spread of dots should provide a more accurate estimation. These scatter plots well depict which method works well and which does not, which agrees well with Table 3. The same holds true for the plots for ImageNet shown in Fig. 5.

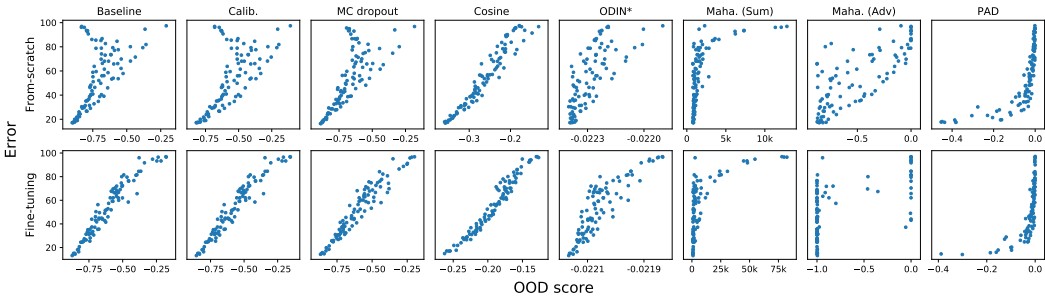

Figure 4: OOD score vs. classification error for $95(=19 \times 5)$ datasets, i.e., $D_o$'s and $D_t$'s (corrupted Food-A images).

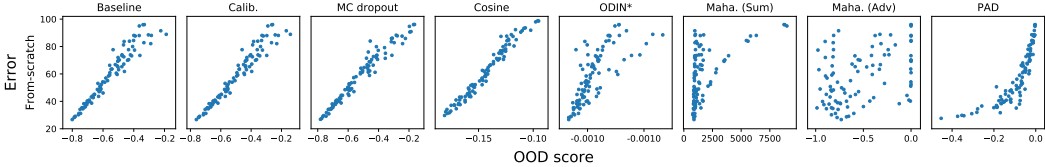

Figure 5: OOD score vs. classification error for $95(=19 \times 5)$ datasets, i.e., $D_o$'s and $D_t$'s (corrupted ImageNet images).

# D   DETECTION OF DOMAIN SHIFT (OFFICE-31)

## D.1   CLASSIFICATION ACCURACY OF THE BASE TASKS

Table 11 shows the classification accuracy of the three networks used by the compared methods for the different domain datasets of Office-31. These networks are trained only on Amazon.

Table 11: The classification accuracy of the three networks trained on Amazon for Amazon, DSLR, and Webcam.

| Dataset | Train-from-scratch | | | Fine-tuning | | |
|---|---|---|---|---|---|---|
| | *Standard* | *MC dropout* | *Cosine* | *Standard* | *MC dropout* | *Cosine* |
| Amazon | 63.0(1.3) | 63.9(2.5) | 67.1(1.8) | 87.8(0.7) | 87.1(0.1) | 87.8(0.3) |
| DSLR | 9.6(0.6) | 7.8(1.6) | 10.0(1.0) | 77.1(1.2) | 78.9(1.3) | 74.2(0.9) |
| Webcam | 7.2(1.9) | 6.8(1.2) | 11.1(0.6) | 73.8(1.3) | 74.1(2.2) | 67.8(0.3) |

## D.2   ADDITIONAL RESULTS

As with the experiments on image corruption, we evaluate how accurately the compared methods can predict the classification error on incoming datasets, $\mathcal{D}_t$'s. Table 4 and Fig. 3 show the error of the predicted classification accuracy and the scatter plots of the OOD score and the true classification accuracy, where $\mathcal{D}_t$'s are created by splitting DSLR and Webcam into sets containing 50 samples. We show here additional results obtained for $\mathcal{D}_t$'s created differently. Table 12 and Fig. 6 show the prediction errors and the scatter plots for $\mathcal{D}_t$'s containing 30 samples. Table 13 and Fig. 7 show those for $\mathcal{D}_t$'s of 100 samples. Table 14 and Fig. 8 show those for using the entire DSLR and Webcam for $\mathcal{D}_t$'s; thus there are only two $\mathcal{D}_t$'s. The standard deviations are computed for $20 \times 3$ trials (20 for random splitting of corruption types for train/val and 3 for network models trained from random initial weights), as explained in Sec. 3.3.3.

Table 12: Errors of the predicted classification error by the compared methods on 30 sample subsets of DSLR and Webcam. The CNN is trained on Amazon and the regressor $f$ is trained using corrupted images of Amazon.

| Method | Train-from-scratch | | Fine-tuning | |
|---|---|---|---|---|
| | MAE | RMSE | MAE | RMSE |
| *Baseline* | 13.3(2.2) | 17.5(2.7) | 10.9(2.0) | 12.8(2.2) |
| *Calib.* | 9.8(2.4) | 12.6(2.7) | 9.9(2.1) | 11.9(2.3) |
| *MC dropout* | 9.7(2.4) | 12.1(3.5) | 10.0(1.6) | 11.7(1.6) |
| *Cosine* | **6.7(1.2)** | **8.3(1.5)** | 9.5(1.8) | 11.5(1.8) |
| *ODIN\** | 8.1(2.4) | 9.3(2.6) | 13.5(3.6) | 16.1(3.8) |
| *Maha. (sum)* | 19.5(3.8) | 22.0(2.9) | **9.1(1.9)** | **11.2(2.0)** |
| *Maha. (adv)* | 33.6(15.7) | 40.0(22.5) | 9.2(1.8) | 11.3(1.9) |
| *PAD* | 13.1(2.9) | 14.7(3.2) | 13.9(2.3) | 16.2(2.3) |

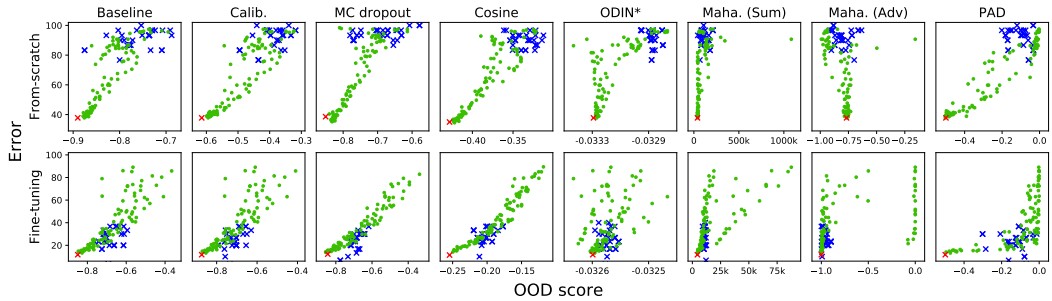

Figure 6: OOD score vs. classification error for $95 (= 19 \times 5)$ $D_o$'s (corrupted Amazon images used for training the regressor $f$; in green), 32 $D_t$'s (subsets of DSLR and Webcam containing 30 samples each; in blue), and $D_s$ (original Amazon images; in red).

Table 13: Errors of the predicted classification error by the compared methods on 100 sample subsets of DSLR and Webcam. The CNN is trained on Amazon and the regressor $f$ is trained using corrupted images of Amazon.

| Method | Train-from-scratch | | Fine-tuning | |
|---|---|---|---|---|
| | MAE | RMSE | MAE | RMSE |
| *Baseline* | 11.1(3.2) | 12.9(3.4) | 10.2(2.7) | 11.0(2.6) |
| *Calib.* | 8.7(3.2) | 10.4(3.7) | 9.2(2.7) | 10.0(2.6) |
| *MC dropout* | 6.9(2.5) | 8.2(2.9) | 8.8(1.9) | 9.7(2.0) |
| *Cosine* | **4.6(1.6)** | **5.5(1.7)** | 7.9(2.1) | 9.0(2.3) |
| *ODIN\** | 6.8(3.1) | 7.3(3.1) | 13.4(3.8) | 15.1(3.7) |
| *Maha. (sum)* | 18.3(4.7) | 19.9(3.9) | **7.4(2.4)** | **8.5(2.4)** |
| *Maha. (adv)* | 33.6(15.3) | 39.1(22.0) | 7.7(2.2) | 8.7(2.2) |
| *PAD* | 8.6(2.3) | 9.3(2.2) | 18.7(3.7) | 19.3(3.7) |

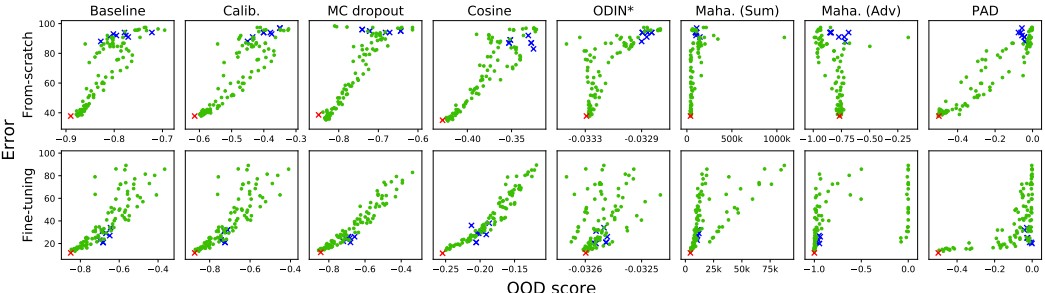

Figure 7: OOD score vs. classification error for $95 (= 19 \times 5)$ $D_o$'s (corrupted Amazon images used for training the regressor $f$; in green), 8 $D_t$'s (subsets of DSLR and Webcam containing 100 samples each; in blue), and $D_s$ (original Amazon images; in red).

Table 14: Errors of the predicted classification error by the compared methods on the entire set of DSLR and Webcam. The CNN is trained on Amazon and the regressor $f$ is trained using corrupted images of Amazon.

| Method | Train-from-scratch | | Fine-tuning | |
|---|---|---|---|---|
| | MAE | RMSE | MAE | RMSE |
| *Baseline* | 9.1(2.1) | 10.4(2.5) | 10.0(2.1) | 10.1(2.1) |
| *Calib.* | 8.3(2.9) | 9.6(3.5) | 9.0(2.3) | 9.0(2.2) |
| *MC dropout* | 6.9(2.9) | 7.8(3.5) | 9.2(1.4) | 9.2(1.4) |
| *Cosine* | **4.0(1.8)** | **4.6(1.8)** | **7.5(2.2)** | **7.5(2.2)** |
| *ODIN\** | 6.5(3.0) | 6.7(3.0) | 12.7(3.5) | 14.1(3.7) |
| *Maha. (sum)* | 17.8(4.8) | 19.2(3.7) | **7.5(1.9)** | 7.6(1.9) |
| *Maha. (adv)* | 27.7(14.0) | 32.9(20.9) | 7.7(1.9) | 7.9(1.9) |
| *PAD* | 6.3(1.9) | 6.5(1.8) | 19.1(2.5) | 19.3(2.6) |

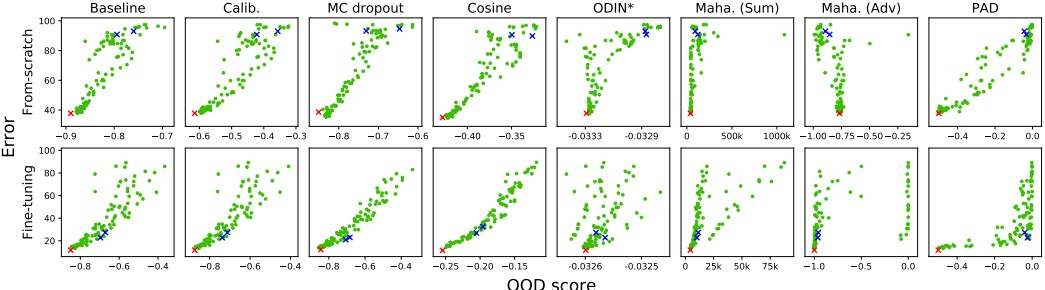

Figure 8: OOD score vs. classification error for $95(= 19 \times 5)$ $D_o$'s (corrupted Amazon images used for training the regressor $f$; in green), 2 $D_t$'s (the entire set of DSLR and Webcam; in blue), and $D_s$ (original Amazon images; in red).

# E  EFFECTIVENESS OF ENSEMBLES

An ensemble of multiple models is known to performs better than MC-dropout we considered in the main experiments for estimation of uncertainty etc. It is also known to be better approximation to Bayesian networks. Thus, we experimentally evaluate ensembles. We consider an ensemble of five models and train each model in two ways, i.e., "from-scratch" and "fine-tuning." We randomly initialize all the weights of each model for the former. We initialize the last layer randomly and other layers with the pre-trained model's weights for the latter. We evaluate ensembles for Baseline and Cosine. Tables 15, 16, 17, and 18 show the results for the three scenarios. In the tables, "(con.)" means confidence is used as an ID score, or equivalently, negative confidence is used as an OOD score. "(en.)" means the entropy is used as an OOD score.

We can observe the following from the tables:

- An ensemble of models performs better than a single model. This is always true for Baseline. The same is true for Cosine except for domain shift detection. (The reason is not clear.)

- An ensemble of Baseline models still performs lower than a single Cosine model for most cases. It sometimes shows better performance for fine-tuned models, but the margin is small.

- Using entropy as OOD score tends to show slightly better performance than using confidence.

We conclude that Cosine's superiority remains true even when we take ensembles into consideration.

Table 15: Irrelevant input detection performance of the ensemble models.

| Method | From-scratch | Fine-Tune |
|---|---|---|
| *Baseline (con.)* | 61.4(12.1) | 97.7(3.1) |
| *Ensemble (con.)* | 67.8(13.7) | 98.3(2.2) |
| *Baseline (en.)* | 64.8(13.7) | 99.2(0.9) |
| *Ensemble (en.)* | 73.4(15.3) | 99.5(0.5) |
| *Cosine* | 83.9(11.4) | 99.0(0.7) |
| *Ensemble cosine* | 85.7(12.9) | 99.1(0.7) |

Table 16: Novel class detection performance of the ensemble models.

| Method | Dog | | Food-A | |
|---|---|---|---|---|
| | From-scratch | Fine-Tune | From-scratch | Fine-Tune |
| *Baseline (con.)* | 61.1(0.8) | 88.7(1.0) | 82.5(0.1) | 84.6(0.2) |
| *Ensemble (con.)* | 64.7 | 89.5 | 84.3 | 86.0 |
| *Baseline (en.)* | 61.6(0.7) | 90.0(1.0) | 83.3(0.1) | 85.4(0.2) |
| *Ensemble (en.)* | 65.7 | 90.8 | 85.0 | 86.8 |
| *Cosine* | 68.8(1.3) | 94.1(0.8) | 83.7(0.1) | 85.7(0.3) |
| *Ensemble cosine* | 72.0 | 94.4 | 85.2 | 86.8 |

Table 17: Errors of the predicted classification error by the ensemble models.

| Method | Food-A (From-scratch) | | Food-A (Fine-tuning) | | ImageNet | |
|---|---|---|---|---|---|---|
| | MAE | RMSE | MAE | RMSE | MAE | RMSE |
| *Baseline (con.)* | 15.8(3.0) | 20.5(3.7) | 6.4(1.3) | 7.9(1.6) | 4.6(0.8) | 6.3(1.0) |
| *Ensemble (con.)* | 12.9(2.3) | 17.1(2.4) | 5.6(1.3) | 7.0(1.8) | 4.0(0.8) | 5.5(1.1) |
| *Baseline (en.)* | 16.8(3.2) | 21.6(3.6) | 6.6(1.0) | 8.4(1.3) | 4.7(0.8) | 6.7(1.1) |
| *Ensemble (en.)* | 14.6(2.0) | 19.3(2.5) | 6.0(0.9) | 7.5(1.3) | 3.9(0.4) | 5.7(0.6) |
| *Cosine* | 6.6(1.3) | 8.2(1.6) | 6.1(1.6) | 7.5(2.2) | 3.8(0.9) | 4.7(1.1) |
| *Ensemble cosine* | 7.3(1.3) | 9.0(1.4) | 6.4(1.6) | 8.0(2.2) | 4.2(1.0) | 5.2(1.3) |

Table 18: Errors of the predicted classification error by the ensemble models on 50 sample subsets of DSLR and Webcam.

| Method | Train-from-scratch | | Fine-tuning | |
|---|---|---|---|---|
| | MAE | RMSE | MAE | RMSE |
| *Baseline (con.)* | 12.1(3.1) | 14.6(3.1) | 10.6(2.3) | 11.7(2.3) |
| *Ensemble (con.)* | 10.4(2.2) | 11.8(2.2) | 9.0(1.1) | 10.9(1.2) |
| *Baseline (en.)* | 11.5(2.5) | 12.7(2.4) | 11.4(2.9) | 13.7(2.9) |
| *Ensemble (en.)* | 11.2(2.2) | 12.6(2.1) | 7.5(1.0) | 8.7(1.1) |
| *Cosine* | 5.6(1.5) | 6.8(1.7) | 8.5(2.0) | 10.0(2.2) |
| *Ensemble cosine* | 5.3(1.1 | 7.1(1.5) | 8.6(1.6) | 10.3(1.8) |

# F ADDITIONAL DETAILS OF EXPERIMENTAL SETTINGS

## F.1 TRAINING OF THE NETWORKS

As is mentioned in the main paper, we employ Resnet-50 in all the experiments. For the optimization, we use SGD with the momentum set to 0.9 and the weight decay set to $10^{-4}$. The learning rate starts at 0.1, and then is divided by 10 depending on the performance of the validation dataset.

To fine-tune a pre-trained network, we use the learning rate of 0.001 for the standard network and that with MC dropout. For the network used with *Cosine*, we use the learning rate of 0.001 to the backbone part and a higher learning rate of 0.1 to the fully-connected layer; the weight decay for the fully-connected layer is set to 0, following Techapanurak et al. (2019) and Hsu et al. (2020).

## F.2 DATASETS

Table 19 shows the specification of the datasets used in our experiments. Note that we modify some of the dataset and use them in several experiments. In the experiments of domain shift detection, we employed image corruption to simulate/model domain shift. The example of the corrupted images are shown in Fig. 9.

Table 19: Specifications of the datasets used in the experiments.

| Dataset | # of classes | # of samples | Ave. image size | Used in |
|---------|-------------|-------------|-----------------|---------|
| Dog Breeds | 120 | 10,222 | $443 \times 387$ | Section 3.1 |
| Plant Seeding | 12 | 5,544 | $357 \times 356$ | Section 3.1 |
| Food-101 | 101 | 101,000 | $496 \times 475$ | Sections 3.1, 3.2, and 3.3.2 |
| CUB-200 | 200 | 11,788 | $468 \times 386$ | Section 3.1 |
| Stanford Car | 196 | 16,185 | $700 \times 483$ | Section 3.1 |
| Oxford-IIIT Pet | 37 | 7,393 | $437 \times 391$ | Section 3.2 |
| ImageNet | 1000 | 1,281,167 | $482 \times 418$ | Section 3.3.2 |
| Office-31 | 31 | 4,110 | $418 \times 418$ | Section 3.3.3 |

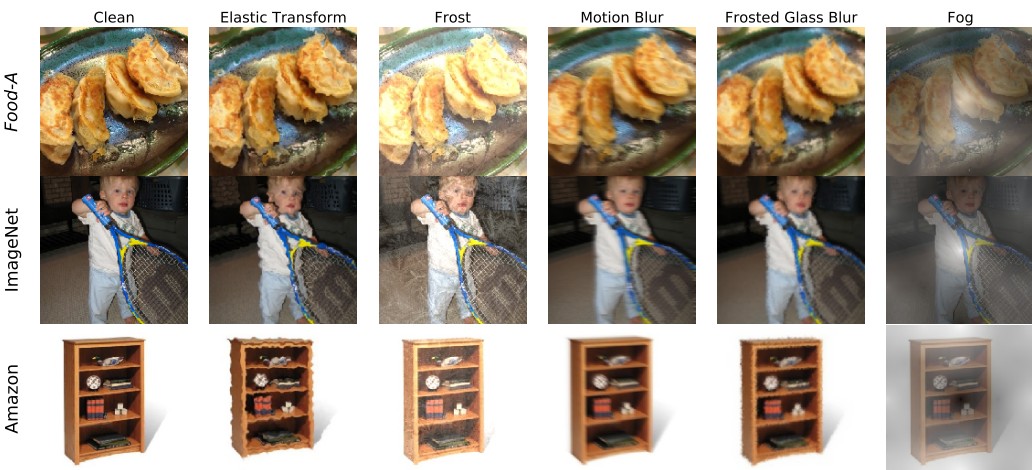

Figure 9: Examples of the corrupted images obtained by applying different types of image corruption (Hendrycks & Dietterich, 2019).

