# OpenReview forum: "Practical Evaluation of Out-of-Distribution Detection Methods for Image Classification"
_ICLR.cc/2021/Conference — Reject_

### Official Review · AnonReviewer2 · 2020-10-27
**Experimental evaluation of OoD methods, but mostly a confirmation of previous work**

**Rating:** 4
**Confidence:** 4

**Review:**

The paper provides an empirical evaluation of various OoD methods . The authors distinguish three different operating scenarios:

+ Irrelevant inputs: in this case, the inputs  simply need to be rejected.
+ Novel classes: detecting inputs that are of the same overall category as that used in training, but from an unseen class of that category.
+ Domain shift, where measurement artifacts and other disturbances cause a shift in the measured p(x) even though the underlying inputs belong to known classes

Multiple methods are compared, using different image datasets to compare OoD detection performance using the AUROC as the evaluation metric.

Pros

+ Paper is reasonably well written, and easy to follow
+ Distinguishing different types of OoD scenarios is a clarifying aspect and useful in this line of work.
+ Shows which methods work under what scenarios.

Cons
+ The biggest concern with this type of work is lack of novelty.  The results are not surprising (as the authors themselves have noted). Fine-tuning/pre-training gives a strong performance boost for OoD detection (as shown in the work of Hendrycks et al 2019) and cosine similarity has been shown to be better than softmax for OoD detection (as shown in the work of Techapanurak et al. (2019); Hsu et al. (2020)).  Further, there is no real discussion or insights into why certain methods work in some scenarios and not others. This results in a rather underwhelming paper for  the reader, and while there are useful takeaways from the paper, as it stands, it does not meet the bar of ICLR acceptance.

References
* Dan Hendrycks, Kimin Lee, and Mantas Mazeika Using pre-training can improve model robustness
and uncertainty. In Proceedings of the International Conference on Machine Learning, 2019.

* Engkarat Techapanurak, Suganuma Masanori, and Takayuki Okatani. Hyperparameter-free out-ofdistribution detection using softmax of scaled cosine similarity. arXiv:1905.10628, 2019.

* Yen-Chang Hsu, Yilin Shen, Hongxia Jin, and Zsolt Kira. Generalized odin: Detecting out-of-distribution image without learning from out-of-distribution data. In Proceedings of the Conference on Computer Vision and Pattern Recognition, 2020.

---

> ### Author Response · Authors · 2020-11-17
> **Response to AnonReviewer2**
>
>
> Q1. The biggest concern with this type of work is lack of novelty. The results are not surprising.
>
> The novelty and contribution of our study are summarized in the following three items. We mention these in the last paragraph of Sec.1 in the revised manuscript.
>
> 1.  Listing three problems that practitioners frequently encounter, we evaluate the existing OOD detection methods on each of them. No previous study has reconsidered OOD detection from such a bird's eye view.
> 2.  Of the three, __domain shift detection has been particularly poorly studied.__ Despite the high demand for its solution in real-world applications, no practical method has been established in the context of deep learning for image recognition. Our study has shown a practical meta-approach, in which many OOD detection methods can be employed as its component, as we did in our experiments.
> 3.  Our experimental evaluation has revealed each method's effectiveness and ineffectiveness on the problems. The biggest surprise is that __the Mahalanobis detector [1] does not perform well for novel class detection or domain shift detection.__ Note that it was ranked among the top performers in the experiments of the previous studies, whose configuration is the most similar to irrelevant input detection in our study, where it performs very well.
>
> Q2. There is no real discussion or insights into why certain methods work in some scenarios and not others.
>
> We have created a new subsection 3.4 “Analyses of the Results,” where we discuss why certain methods work in some scenarios and not others. It includes the following discussion.
> * While it works very well on irrelevant input detection, the Mahalanobis detector does not perform well on novel class detection and domain shift detection.
> 	* __Its rugged performance on novel class detection is attributable to the inaccuracy of modeling ID sample distribution in the feature space.__ It fits a Gaussian distribution to ID samples belonging to each class and uses the same covariance matrix for all the classes. This strategy might work well on easy cases when incoming OOD samples are mapped distantly from the Gaussian distributions. However, __such a simple modeling method will not work in more challenging cases.__ For instance, incoming OOD samples could be mapped near the ID distributions, as in novel class detection. In such cases, the ID sample distribution needs to be very precisely modeled, for which the assumption of Gaussian distributions with a single covariance matrix is inadequate.
> 	* The Mahalanobis detector does not work well on domain shift detection, either. Differently from the above, __this is attributable to the mismatch between the problem's nature and the method' design.__ By definition, domain shift detection itself is not OOD detection; it is a problem of detecting the performance deterioration of a classifier on novel inputs, or more simply, predicting the accuracy of the classifier. This explains well the good performance of all the confidence-based methods in our experiments. However, the Mahalanobis detector is more like an anomaly detection method, although the authors' paper shows its similarity with a softmax classifier. __An input sample for which the network can make a correct classification can be detected as an `anomaly' by the Mahalanobis detector.__
>
> Reference:
> [1] Kimin Lee, Kibok Lee, Honglak Lee, and Jinwoo Shin. A simple unified framework for detecting out-of-distribution samples and adversarial attacks. In Advances in Neural Information Processing Systems, 2018.

---

### Official Review · AnonReviewer3 · 2020-10-28
**good evaluation of practical OOD detection, only minor issues**

**Rating:** 8
**Confidence:** 5

**Review:**

#### Summary
This paper is about a comparison of methods for out of distribution detection on image classification. The authors compare along three different lines: Irrelevant inputs, novel classes, and domain shift. The results show that cosine similarity consistently outperforms other methods across all sub-tasks.

The comparison made in this work is novel, as most OOD detection methods are evaluated on standard academic datasets (CIFAR10/100, SVHN, etc), which could hide issues that are presented in more realistic scenarios.


#### Reasons for Score
I believe that this paper touches a subject of importance to practitioners and researchers, the performance of OOD detection methods in more realistic settings and with three different sub-tasks inside the OOD detection task.
The evaluations are made correctly, in a variety of datasets, ensuring the robustness of the conclusions that were made.
My only concerns with this paper are the lack of comparisons with ensembles, which could change which method works best.


#### Pros
- The evaluation is comprehensive, with a variety of datasets, properly selected, ensuring robust results and conclusions. There is also a variety of tasks inside out of distribution detection (new inputs, novel classes, and domain shift).
- The evaluation produces strong conclusions, with one method (cosine similarity) clearly outperforming the rest. This provides good evidence for practitioners to select OOD detection methods in the future.
- The evaluation is made with a practical point of view, motivated by real-world examples, that differ from purely academic benchmarks. This is very useful for practitioners.
- A nice appendix that provides detailed results that can be used for future comparison.


#### Cons
- I believe there is a missing reference that is very relevant for this work, namely Ovadia et al. "Can You Trust Your Model’s Uncertainty? Evaluating
Predictive Uncertainty Under Dataset Shift" NeurIPS 2019. Many methods evaluated in Ovadia et al. like ones relying on variational inference are not evaluated in this work, so I suggest to take a look and include additional baselines. Note that Ovadia et al. also defines corruption methods for evaluating dataset shift.
- I think there are missing methods to compare, for example ensembles. The paper mentions that MC-Dropout "is an approximation of BNNs that is computationally more efficient than en ensemble of networks". I do not think this is true, ensembles and MC-Dropout are equally slow in predicting a forward pass, but an ensemble usually performs better in OOD detection and produces higher quality uncertainty (see Lakshminarayanan et al. 2017, Figure 3 and Ovadia et al. 2019). If MC-Dropout is used for comparison, then Ensembles should also be used since generally ensembles outperform MC-Dropout in terms of uncertainty quality. Note that this might change which method works better, or even create new baselines, as ensembling can be combined with cosine similarity OOD detection.


#### Questions for Rebuttal Period
Why were ensembles not considered as one of the evaluated methods?


#### Minor Issues
- In Section 3.1, please mention on which dataset are models pre-trained, I assume it is on ImageNet, but this needs to be clearly specified.
- Please mention in the Appendix that number of forward passes used in MC-Dropout
- I think method should be separated into two groups, one where OOD detection capabilities are given by the quality of uncertainty (like Bayesian NNs, MC-Dropout, Ensembles, etc), and other methods where an score is produced specifically for OOD detection and are not direct measures of uncertainty (not probabilities). This could shed a light on why methods work so differently.
- Sections 2.2 and 3.3.1 could be rewritten to clearly describe the OOD score produced by each method, for example in some cases the ID score is describe, how is the OOD score derived in these cases? This is important since an OOD score is used in Section 3.3.
- For probability-based methods like MC-Dropout, entropy of the output probabilities can also be used as an OOD score.
- In Figure 3, the x labels for Maha (Sum) are too close to each other and might hinder reading, I recommend to rewrite as 25K/50K/75K or rotate them a bit.
- For more clarity and a self-contained work, the authors could add a summary description of each method in mathematical terms to the appendix.

---

> ### Author Response · Authors · 2020-11-19
> **Response to AnonReviewer3**
>
>
> Q1. If MC-Dropout is used for comparison, then Ensembles should also be used since generally ensembles outperform MC-Dropout in terms of uncertainty quality. Note that this might change which method works better, or even create new baselines, as ensembling can be combined with cosine similarity OOD detection.
>
> We did additional experiments to test ensembles. We consider an ensemble of five models and train each model in two ways, i.e., “from-scratch” and “fine-tuning.” We randomly initialize all the weights of each model for the former. We initialize the last layer randomly and other layers with the pre-trained model’s weights for the latter. __We evaluate ensembles for Baseline and Cosine.__ The following tables show the results for the three scenarios. In the tables, “(con.)” means confidence is used as an ID score, or equivalently, negative confidence is used as an OOD score. “(en.)” means the entropy is used as an OOD score.
>
> We can observe the following from the tables:
> * An ensemble of models performs better than a single model. This is always true for Baseline. The same is true for Cosine except for domain shift detection. (The reason is unknown.)
> * __An ensemble of Baseline models still performs lower than a single Cosine model for most cases.__ It sometimes shows better performance for fine-tuned models, but the margin is small.
> * Using entropy as OOD score tends to show slightly better performance than using confidence.
>
> We conclude that Cosine’s superiority remains true even when we take ensembles into consideration. Although an ensemble shows slightly better performance in a few cases, we think it is better to treat ensembles differently from an application point of view, because i) ensembles are not efficient in terms of memory space and computational time, which is also the reason why we employ MC-Dropout, and ii) we can employ an ensemble with most of the compared methods, which is expected to improve performance; thus, we can regard using an ensemble as an option.
>
> __Irrelevant input detection:__
>
> ||From-scratch|Fine-tuning|
> |:-:|:-:|:-:|
> |Baseline (con.)|61.4(12.1)|97.7(3.1)|
> |Ensemble (con.)|67.8(13.7)|98.3(2.2)|
> |Baseline (en.)|64.8(13.7)|99.2(0.9)|
> |Ensemble (en.)|73.4(15.3)|99.5(0.5)|
> |Cosine|83.9(11.4)|99.0(0.7)|
> |Ensemble (cos)|85.7(12.9)|99.1(0.7)|
>
> __Novel class detection:__
>
> ||Dog|Dog|Food-A|Food-A|
> |:-:|:-:|:-:|:-:|:-:|
> ||From-scratch|Fine-tuning|From-scratch|Fine-tuning|
> |Baseline (con.)|61.1(0.8)|88.7(1.0)|82.5(0.1)|84.6(0.2)|
> |Ensemble (con.)|64.7|89.5|84.3|86.0|
> |Baseline (en.)|61.6(0.7)|90.0(1.0)|83.3(0.1)|85.4(0.2)|
> |Ensemble (en.)|65.7|90.8|85.0|86.8|
> |Cosine|68.8(1.3)|94.1(0.8)|83.7(0.1)|85.7(0.3)|
> |Ensemble (cos)|72.0|94.4|85.2|86.8|
>
> __Domain shift detection:__
>
> ||Food-A|Food-A|Food-A|Food-A|ImageNet|ImageNet|
> |:-:|:-:|:-:|:-:|:-:|:-:|:-:|
> ||From-scratch|Fine-tuning|From-scratch|Fine-tuning|From-scratch|Fine-tuning|
> ||MAE|RMSE|MAE|RMSE|MAE|RMSE|
> |Baseline (con.)|15.8(3.0)|20.5(3.7)|6.4(1.3)|7.9(1.6)|4.6(0.8)|6.3(1.0)|
> |Ensemble (con.)|12.9(2.3)|17.1(2.4)|5.6(1.3)|7.0(1.8)|4.0(0.8)|5.5(1.1)|
> |Baseline (en.)|16.8(3.2)|21.6(3.6)|6.6(1.0)|8.4(1.3)|4.7(0.8)|6.7(1.1)|
> |Ensemble (en.)|14.6(2.0)|19.3(2.5)|6.0(0.9)|7.5(1.3)|3.9(0.4)|5.7(0.6)|
> |Cosine|6.6(1.3)|8.2(1.6)|6.1(1.6)|7.5(2.2)|3.8(0.9)|4.7(1.1)|
> |Ensemble (cos)|7.3(1.3)|9.0(1.4)|6.4(1.6)|8.0(2.2)|4.2(1.0)|5.2(1.3)|
>
>
> ||Amazon|Amazon|Amazon|Amazon|
> |:-:|:-:|:-:|:-:|:-:|
> ||From-scratch|Fine-tuning|From-scratch|Fine-tuning|
> ||MAE|RMSE|MAE|RMSE|
> |Baseline (con.)|12.1(3.1)|14.6(3.1)|10.6(2.3)|11.7(2.3)|
> |Ensemble (con.)|10.4(2.2)|11.8(2.2)|9.0(1.1)|10.9(1.2)|
> |Baseline (en.)|11.5(2.5)|12.7(2.4)|11.4(2.9)|13.7(2.9)|
> |Ensemble (en.)|11.2(2.2)|12.6(2.1)|7.5(1.0)|8.7(1.1)|
> |Cosine|5.6(1.5)|6.8(1.7)|8.5(2.0)|10.0(2.2)|
> |Ensemble (cos)|5.3(1.1|7.1(1.5)|8.6(1.6)|10.3(1.8)|
>
> Q2. I believe there is a missing reference that is very relevant for this work, namely Ovadia et al. "Can You Trust Your Model’s Uncertainty? Evaluating Predictive Uncertainty Under Dataset Shift" NeurIPS 2019. Many methods evaluated in Ovadia et al. like ones relying on variational inference are not evaluated in this work, so I suggest to take a look and include additional baselines. Note that Ovadia et al. also defines corruption methods for evaluating dataset shift.
>
> Thank you for the suggestion. We mention the paper in the revised version.
>
> Q3. In Section 3.1, please mention on which dataset are models pre-trained, I assume it is on ImageNet, but this needs to be clearly specified.
>
> We explain it in Section 3.
>
> Q4. Please mention in the Appendix that number of forward passes used in MC-Dropout.
>
> We add an explanation to the main text of the revised version.

---

> > ### Author Response · Authors · 2020-11-19
> > **Response to AnonReviewer3 (cont.)**
> >
> >
> > Q5. I think method should be separated into two groups, one where OOD detection capabilities are given by the quality of uncertainty (like Bayesian NNs, MC-Dropout, Ensembles, etc), and other methods where an score is produced specifically for OOD detection and are not direct measures of uncertainty (not probabilities). This could shed a light on why methods work so differently.
> >
> > We modify the result tables in the revised version with the explanation in Section 3.2.
> >
> >
> > Q6. Sections 2.2 and 3.3.1 could be rewritten to clearly describe the OOD score produced by each method, for example in some cases the ID score is describe, how is the OOD score derived in these cases? This is important since an OOD score is used in Section 3.3.
> >
> > We have revised the explanation.
> >
> > Q7. For probability-based methods like MC-Dropout, entropy of the output probabilities can also be used as an OOD score.
> >
> > As shown in the above experiments, entropy often yields better results. We chose confidence (i.e., the max softmax probability) as primary score, following Hendrycks et al. 2017 [1].
> >
> > Reference:
> >
> > [1] Dan Hendrycks and Thomas Dietterich. Benchmarking neural network robustness to common corruptions and perturbations.  In Proceedings of the International Conference on Learning Representations, 2019.

---

### Official Review · AnonReviewer1 · 2020-10-28
**Important problem but the paper is not ready**

**Rating:** 3
**Confidence:** 3

**Review:**

##########################################################################

Summary:

The paper empirically analyzes the evaluation framework of the current OOD detection systems for the image recognition task, specifically the evaluation described in [1] using Max-softmax and calibrated confidence. They motivate the paper by the necessity of having better evaluation for OOD detection to be reflective of real world scenarios. The addressed problem is interesting and valuable for the field as many of the defined OOD datasets, and evaluation metrics may not cover many real-world scenarios. They specifically addressed three scenarios, inputs that i) are irrelevant to the task ii) are from novel classes and iii) are from another domain (domain shift), which for the first 2 scenarios, they only evaluate them as unseen classes and not distinguish between them. Based on my understanding of the paper, they compare 5 OOD detection methods from the literature, suggest a few test datasets/scenarios and conclude using cosine similarity is consistently favorable for evaluation, and the choice of using confidence-based methods in case of domain shift detection scenarios.

##########################################################################
[Q]: questions, and comments on the paper. [S] strength [W] weakness

Q1: Questions about the choices of the datasets:

1.1. What are the reasoning for the choices of datasets?
1.2. Why are these datasets considered reflective of real world scenarios? For example, why didn’t you analyze the models on some OOD datasets like Imagenet-A, -O, -P.

Q2: Could you elaborate more on the specific contributions of the paper?

Q3: What does “genuine OOD samples” mean in section 2.2?

Q4: There is no related work part In the paper.

Q5: Many recent and relevant papers are not cited or compared against. As an example, it is mentioned in the Domain shift detection [2] study in NLP, that is not the focus of the paper. However, there are multiple recent works in Computer Vision also addressing the domain or distributional shift in vision scenarios. In particular, for the example surveillance system scenarios mentioned in page 2, there are datasets and papers addressing the domain shift that should be cited such as [3 (a review paper), 4] or [5].

Q6: The following statement from the paper is inaccurate as a general statement. Which kind of pre-training contributes to performance improvement? In addition, there are many scenarios where pre-training or fine-tuning lead to reduced performance on seen classes or on the original data distribution.
“(i) Using a pre-trained network always contributes to performance improvements, confirming the study of Hendrycks et al [1]. (2019). ”

Q7: For the evaluation using fine-tuning, the impact of the results on the seen classes (the current classes in the dataset), are not re-evaluated. In many cases, fine-tuning in a new distribution, contributes to reduced performance in seen classes or former distribution. I recommend checking the literature of generalized zero-shot learning and specifically the use of harmonic mean as a measurement.

##########################################################################
Reason for the score:

S1: The paper is well-motivated and addresses an interesting problem.

S2: I believe there is room for improvement.

W1: The contributions of the paper are not clear. The ablation studies to compare the approaches are not 	sufficient.

W2: I am not convinced the current set up can be sufficient for generalizable decisions on choice of evaluations.

W3: Choices of the datasets for comprehensive guidance is unclear.

W4: There is no related work part  in the paper and multiple recent and relevant papers are not being compared against or cited.

##########################################################################
References:
[1] Hendrycks and Kevin Gimpel, A baseline for detecting misclassified and out-of-distribution examples in neural networks, ICLR 2017

[2] Elsahar et al, To annotate or not? predicting performance drop under domain shift, IJCNLP 2019.

[3] Toldo et al, Unsupervised Domain Adaptation in Semantic Segmentation: a Review, CVPR 2020

[4] Zhang et al, Curriculum Domain Adaptation for Semantic Segmentation of Urban Scenes, CVPR 2017.

[5] Ganin et al, Unsupervised Domain Adaptation by Backpropagation, ECCV 2016.

---

> ### Author Response · Authors · 2020-11-17
> **Response to AnonReviewer1**
>
>
> Q1. What is the reasoning for the choices for datasets? Why not ImageNet-A, -O, -P?
>
> We believe the current choice of datasets are good representatives of real-world scenarios. From our experience, we don’t think we will get a new finding, if we add more dataset. To prove this, we conducted experiments on ImageNet-A and -O for irrelevant input detection; see the table below. We can see our conclusion holds true for the new datasets. In summary, we think our experiments provide sufficient clues for practitioners to deal with their problems.
>
> ||From-scratch|From-scratch|Fine-tuning|Fine-tuning|
> |-|:-:|:-:|:-:|:-:|
> ||Five Datasets|ImageNet-A,-O|Five Datasets|ImageNet-A,-O|
> |Baseline|61.4(12.1)|80.6(9.6)|97.7(3.1)|90.7(11.5)|
> |Calib.|66.7(12.8)|81.0(11.0)|95.6(6.4)|91.5(11.5)|
> |MC dropout|66.3(10.6)|84.1(7.9)|98.2(3.2)|91.7(9.6)|
> |ODIN*|73.3(10.1)|77.5(14.5)|98.6(1.6)|91.6(11.8)|
> |Cosine|83.9(11.4)|96.1(2.1)|99.0(0.7)|97.3(4.0)|
> |Maha. (sum)|76.4(9.7)|90.8(12.4)|99.0(1.0)|97.7(2.7)|
> |Maha. (adv)|56.2(18.6)|78.1(18.8)|98.1(2.1)|92.1(10.2)|
>
> Q2. Could you elaborate more on the specific contributions?
>
> The novelty and contribution of our study are summarized in the following three items. We mention these in the last paragraph of Sec.1 in the revised manuscript.
>
> 1.  Listing three problems that practitioners frequently encounter, we evaluate the existing OOD detection methods on each of them. No previous study has reconsidered OOD detection from such a bird's eye view.
> 2.  Of the three, __domain shift detection has been particularly poorly studied.__ Despite the high demand for its solution in real-world applications, no practical method has been established in the context of deep learning for image recognition. Our study has shown a practical meta-approach, in which many OOD detection methods can be employed as its component, as we did in our experiments.
> 3.  Our experimental evaluation has revealed each method's effectiveness and ineffectiveness on the problems. The biggest surprise is that __the Mahalanobis detector [1] does not perform well for novel class detection or domain shift detection.__ Note that it was ranked among the top performers in the experiments of the previous studies, whose configuration is the most similar to irrelevant input detection in our study, where it performs very well.
>
> Q3. What is “genuine OOD samples”?
>
> By the term, we intended to mean “real OOD samples that we will encounter at test time”. It is unrealistic to assume their availability at training time. We have revised the part.
>
> Q4. No related work part in the paper.
>
> We have created a new section “Related Work” in the revised manuscript using one additional page.
>
> Q5. Many recent and relevant papers are not cited or compared against… multiple recent works also addressing the domain or distributional shift in vision scenarios. In particular, for the example surveillance system scenarios mentioned in page 2, there are datasets and papers addressing the domain shift that should be cited such as [3,4,5].
>
> We are aware of a considerable number of studies of domain adaptation in the field. However, __these studies assume the domain of each dataset to be already known.__ Their focus is on how to adapt a model trained on a domain to another domain. As far as we know, __no study in the field of computer vision has considered the problem of detecting when the domain has shifted.__ This also applies to the suggested papers [3,4,5]. We have added this discussion to the revised manuscript.
>
> Q6. The sentence is not always true, i.e., “(i) Using a pre-trained network always contributes to performance improvements, confirming the study of Hendrycks et al [1]. (2019).”.
>
> It is not intended as a general statement on the use of pre-trained models. It is only a summary of our experiments. We have revised the statement.
>
> Q7. For the evaluation using fine-tuning, the impact of the results on the seen classes are not re-evaluated.
>
> The argument seems to be about continual learning. It is beyond the scope of this study.
>
> Reference:
> [1] Kimin Lee, Kibok Lee, Honglak Lee, and Jinwoo Shin. A simple unified framework for detecting out-of-distribution samples and adversarial attacks. In Advances in Neural Information Processing Systems, 2018.

---

### Official Review · AnonReviewer4 · 2020-10-29
**Many experiments, not enough analysis/insights**

**Rating:** 4
**Confidence:** 4

**Review:**

Summary:
----------------
The goal of this paper is to evaluate methods for detecting out-of-distribution samples in a more comprehensive fashion than prior work. To this end, it distinguishes between three different cases of OOD samples:

(1) detecting irrelevant inputs: defined in this case as being "out of concern" for a task of interest, e.g. when classifying dogs is the task of interest, irrelevant inputs would be images of food, plants, birds, and cars

(2) detecting novel classes: e.g. if the task of interest is classifying 15 types of dog, a novel class would be an unseen breed of dog

(3) detecting domain shift: that is detecting samples that belong to a known class but stem from a different generating distribution due to image corruptions (a la Hendrycks &  Dietterich, 2019)  or a different imaging setup/setting (a la Saenko et al., 2010)

An impressive number of experiments are carried out in which the same set of OOD detection methods are compared across these different scenarios. From the results, it is concluded that using a pre-trained network improves OOD detection (in line with previous results), and that not all methods perform well across the different settings, but one particular method does: cosine similarity (Techapanurak et al., 2019; Hsu et al., 2020).

Strengths & Weaknesses:
----------------------------------
Given the dizzying amount of papers continually being released in any ML subfield, any attempt to step back and evaluate recent methods on even footing are always welcome in my book and can provide more value than adding yet another method to the literature. I appreciate the effort to distinguish between different cases of OOD detection as well as the amount of experiments with a good selection of methods including some very recent ones.

Despite the above, I have reservations about this paper:

While there are a lot of experiments, the novel conclusions are sparse and very little space is afforded for a discussion of these: a single paragraph on page 8. It's a bit difficult to see the forest from the trees.

Looking beyond the individual results, the key insight here to me is that there are different kinds of OOD detection and that it's important to consider these different settings when evaluating a method. However, Hsu et al. (CVPR '20) also distinguish between two settings in their evaluation: semantic and non-semantic shift. These results are unfortunately not discussed in this submission. To be clear, here the distinction is a bit more nuanced. The first two settings (1) "irrelevant inputs" and (2) "novel classes" correspond to different kinds of semantic shift (the latter to fine-grained classes, e.g. dog breeds), and the last setting "domain shift" corresponds to non-semantic shift. While I agree that (1) and (2) are real and distinct practical scenarios, I don't think either implies a specific degree of "semantic closeness". What is "novel" and what is "irrelevant" is application-dependent. I would not tie "novel" and "irrelevant" to certain sets of classes, but instead focus on the effect of "semantic closeness" on OOD performance as a more nuanced way of looking at semantic shift.

Now a detailed discussion of the experiments themselves:

The experiments on "detection of irrelevant inputs" (3.1) confirms results from Hendrycks et al. (2019) that using a pre-trained network is better for OOD detection than training from scratch. Given the choice of datasets, the OOD detection is almost perfect (above 95%) except when the in-distribution is the food dataset. This leads me to question the choice of datasets, because I'm certain that OOD detection remains a difficult problem these almost perfect results notwithstanding.

The experiments on "detection of novel classes" (3.2) find that within the food dataset, four methods perform almost identically well (Baseline, Calib, MC dropout, Cosine), and outperform ODIN* and Mahalanobis. The conclusion is similar from the Dog experiment, except that Cosine performs the best by some margin. This contradicts the results in Hsu et al. 2020, where it is shown that both ODIN* and a variant of Mahalanobis significantly outperform Baseline across different settings. Here it is chalked up to the similarity between in-distribution and out-distribution images.

There are also a couple of problems with the experimental setting: The dataset of dog images contains 25 classes, some of which overlap with the Imagenet classes. This violates the OOD aspect. The authors explain that they "intentionally leave this overlap to simulate a similar situation that could occur in practice", but this really isn't convincing. In the case of the food dataset on the other hand, "eight classes [out of 101] contained in the ImageNet dataset" are removed. Perhaps it is inconvenient to retrain an ImageNet classifier without the offending dog classes, but then maybe another dataset should have been selected. This dataset contamination was not discussed in section 3.1, despite the use of the same dataset. Was this not an issue?

The final set of experiments on "detection of domain shift" (3.3) were confusing to me. I found the description of the problem formulation as well as the experimental setting (3.3.1, first part of 3.3.2) hard to followm especially the second paragraph of 3.3.1. Here also the goal is now to predict the classification error from the OOD scores, which departs from the setting considered for the first two sets of experiments. I would recommend cutting out section 3.3.2 and spending more time on analysing the results as I don't see what these experiments add to the paper.

Also, a new method is suddenly introduced (PAD*) and I'm not sure why. The way it's trained also appears to contradict the following statement in the introduction "we follow the argument shared by many recent studies [...] that OOD detection methods should not assume the availability of explicit OOD samples at training time". PAD* involves "train[ing] a binary classifier using portions of D_s and D_t to distinguish the two".

Conclusion:
----------------

A lot of effort went into conducting a large number of experiments with the goal of comparing several OOD detection methods across different scenarios. While I appreciate this effort and agree that it makes sense to consider these different scenarios when comparing methods, I don't think there are sufficiently novel insights to warrant acceptance with very little discussion of the results.

Evaluating OOD detection on fine-grained vs. non-fine-grained classes is interesting, and I would have liked to see more exploration of this. Unfortunately, the choice of datasets for these experiments makes the task appear to be too easy in one case, and in the other case is not truly OOD due to overlap with ImageNet classes.

Given that this is an evaluation of existing methods with little analysis, I would have been more inclined to accept if this paper established a convincing benchmark that could be easily adopted by researchers in this area, but due to the aforementioned problems this is not really the case. Perhaps one could consider DomainNet (Peng et al. ICCV '19) as a dataset that would support such an analysis at different semantic levels together with the non-semantic component. Additionally, one could add the image corruptions proposed by Hendrycks & Dietterich.

---

> ### Author Response · Authors · 2020-11-17
> **Response to AnonReviewer4**
>
>
> Before answering all these questions, let us state our motivation behind this study. While tons of methods have been proposed so far in the fields of deep learning and computer vision, we have felt, as scholars in an academic field, that __academicians have not responded well to serious demands for help from practitioners.__ This is true for OOD detection and domain shift detection. We list three application scenarios, for each of which we show which method works and which not. __We believe our study will offer the first guide of this kind for practitioners.__ It includes how OOD detection methods can be applied to domain shift detection, for which no established method exists in the context of deep learning for image classification.
>
> Q1. Skepticism on our classification of OOD samples into irrelevant inputs and novel classes. Discussion of its relation to Hsu et al.’s classification method is missing.
>
> In our understanding, the reviewer’s comments are summarized as follows: The classification into irrelevant inputs and novel classes are arbitrary. The reviewer suggests we use “semantic closeness” between ID and OOD samples to classify or sort OOD samples. __We disagree with this suggestion, simply because we do not think it is feasible.__ For instance, we don’t know how to define or measure “semantic closeness.” We would instead agree that “what is novel and what is irrelevant is application-dependent.” While the reviewer views it negatively, we think it is rather reasonable. We do not think it is possible to characterize OOD samples without specifying applications.
>
> One may view our classification as somewhat arbitrary and vague. Unfortunately, Hsu et al. [1] 's classification scheme does not provide help. Their method is summarized as follows: OOD samples are classified into i) samples of a novel class but in a known domain, called semantic shift (S), and ii) samples of known class in a novel domain, called non-semantic shift (NS). An issue with this classification is that there is arbitrariness in “what constitutes a different domain.” For instance, __the reviewer classifies both irrelevant inputs and novel classes to S. We would classify the former to S+NS because we think dog images and plant images are from different domains__; both being “real images” does not mean they are from the same domain. We added these discussions to the newly added Sec.4.
>
> Based on this consideration, we have chosen the datasets in the two scenarios from a general application perspective. The results have shown that different methods show different performances in the two scenarios. We believe that practitioners will gain useful clues.
>
> Q2. The final set of experiments on "detection of domain shift" (3.3) were confusing to me. … Here also the goal is now to predict the classification error from the OOD scores, which departs from the setting considered for the first two sets of experiments.
>
> As the reviewer recognizes, domain shift detection is precisely not identical to OOD detection. Why we consider it in this paper is that OOD detection methods can be applied to it. (Note that our paper title is “Practical Evaluation of Out-of-distribution Methods …”) Moreover, __despite many demands from practitioners, there is no established method for domain shift detection in the context of deep learning for image classification.__ In that sense, one may think that __Sec.3.3 includes a new proposal__, i.e., a novel meta-approach to the problem, in which any OOD detection method can be used as its component. As the manuscript lacked these explanations, we added the above explanation to the 5th paragraph of Sec.1.
>
> Q3. (The results with a pretrained model on irrelevant input detection are almost perfect.) This leads me to question the choice of datasets, because I’m certain that OOD detection remains a difficult problem these almost perfect results notwithstanding.
>
> A. We do not think our choice of datasets is inadequate. Could you tell us the specific datasets that are more suitable for the evaluation? Please note that our results for irrelevant input detection agree well with those reported in many previous studies. Its choice of datasets is the most similar to the standard benchmark test employed in the previous studies (except the difference in image resolution), as we mention in the manuscript.
>
> Reference:
> [1] Yen-Chang  Hsu,  Yilin  Shen,  Hongxia  Jin,  and  Zsolt  Kira.   Generalized  odin:  Detecting  out-of-distribution image without learning from out-of-distribution data.  In Proceedings of the Conference on Computer Vision and Pattern Recognition, 2020.

---

> > ### Author Response · Authors · 2020-11-17
> > **Response to AnonReviewer4 (cont.)**
> >
> >
> > Q4. (The results on novel class detection show that Baseline, Calib, MC dropout, and Cosine outperform ODIN* and Mahalanobis.)This contradicts with the results in Hsu et al. 2020.
> >
> > There is no contradiction. The difference is fully explainable by the difference in the datasets used. While ID and OOD are distinct in irrelevant input detection, ID and OOD are close to each other in novel class detection. We can observe the same tendency as Hsu et al. [1] 's results in our results of irrelevant input detection.
> >
> > Q5. The dataset of dog images contains 25 classes, some of which overlap with the Imagenet classes. This violates the OOD aspect.
> >
> > We do not think having some overlap with ImageNet classes is meaningless, although it may violate the strict OOD definition. On the contrary, __it is essential to examine such cases because finetuning an ImageNet pre-trained model is the gold standard__ for training a network on all sorts of new tasks. As long as the target is natural image classification, it is implausible that there is precisely no overlap between OOD samples (or ID samples) and ImageNet classes. Even if class names differ, it does not mean overlap-free (e.g., snakes and pythons). Moreover, it is easy to expect that a pre-trained model trained on a bigger dataset with a greater number of classes will be available in the future. Then, avoiding formal overlaps of the class names will be increasingly meaningless. Thus, if it violates the OOD definition, then the definition lacks practical usefulness. Again, the most important is how well the experiments reflect real-world problems.
> > Having said that, __there are several cases where there is strictly no overlap__ with the classes learned by the pre-trained model. __Our experiments on Food-A cover such cases.__ Finally, it should be noted that the results of Dogs and those of Food-A give us an identical conclusion; the ranking of the compared methods are mostly the same. This consistency reinforces the validity of our experimental configuration.
> >
> > Q6. This dataset contamination was not discussed in section 3.1, despite the use of the same dataset. Was this not an issue?
> >
> > Our answer is the same as above. Moreover, if that is considered meaningless, so are many previous studies. Additionally, Table 6 in the appendix shows more detailed results containing detection performance for each OOD dataset, from which one can assess the impact of the closeness between the OOD samples to ImageNet classes on detection accuracy. However, we do not see any impact that changes the methods' ranking.
> >
> > Q7. Also, a new method is suddenly introduced (PAD*) and I'm not sure why. The way it's trained also appears to contradict the following statement in the introduction… PAD* involves "train[ing] a binary classifier using portions of D_s and D_t to distinguish the two".
> >
> > Moreover, as mentioned above, domain shift detection for image classification is a forgotten problem in the community, despite its practical importance; there is no established method. Our solution is to tackle the problem from the viewpoint of OOD detection. As it is not the only solution, we evaluate PAD* for comparison, which is reported in [2] to perform well in the field of NLP. It is the only study we have found that considers domain shift detection in the context of deep learning.
> >
> > References:
> >
> > [1] Yen-Chang  Hsu,  Yilin  Shen,  Hongxia  Jin,  and  Zsolt  Kira.   Generalized  odin:  Detecting  out-of-distribution image without learning from out-of-distribution data.  In Proceedings of the Conference on Computer Vision and Pattern Recognition, 2020.
> >
> > [2] Hady Elsahar and Matthias Galle. To annotate or not? predicting performance drop under domain shift. In Proceedings of the Conference on Empirical Methods in Natural Language Processing and the International Joint Conference on Natural Language Processing, 2019.

---

### Decision · Program_Chairs · 2021-01-07
**Final Decision**

**Decision:**

Reject

**Comment:**

This paper proposes an OOD evaluation framework under three categories: irrelevant input detection, novel class detection, and domain shift detection. As with several reviewers, the AC recognizes the importance and effort to distinguish between different cases of OOD detection, as well as the amount of experimental comparison across several prominent methods in literature (MSP, MC-dropout, cosine similarity, ODIN, Mahalanobis).

Despite being well-motivated, three knowledgeable reviewers find the paper not ready yet for publication at ICLR. The AC recommends a rejection, given the standing major concerns from the reviewers. The AC is hopeful that the paper can be significantly improved by

- sufficiently discussing and highlighting the novel insights of the results.
- a more rigorous definition of  "novel" vs. "irrelevant" inputs. There seem to be overlapping definitions between what Hsu et al. considered vs. this paper. In particular,  Hsu et al distinguish i) samples of a novel class but in a known domain, called semantic shift (S), and ii) samples of known class in a novel domain, called non-semantic shift (NS), both of which are reconsidered in this paper. Therefore, the novelty of this submission is more precisely to distinguish within the category of semantic shift. The AC agrees that this might deem some more rigorous measurement and definition of "semantic closeness".
- The AC also finds the evaluation of domain shift in Section 3.3.2 may be potentially misleading the community, as it falls out of the standard OOD scope. The notion of common corruption is closer to the robustness problem (which is how ML model predictions changes w.r.t some delta changes in the input space). The changes may not be substantial enough to be "out-of-distribution".